# ES-Based Jacobian Enables Faster Bilevel Optimization

## Abstract

Bilevel optimization (BO) has arisen as a powerful tool for solving many modern machine learning problems. However, due to the nested structure of BO, existing gradient-based methods require second-order derivative approximations via Jacobian- or/and Hessian-vector computations, which can be very costly in practice, especially with large neural network models. In this work, we propose a novel BO algorithm, which adopts Evolution Strategies (ES) based method to approximate the response Jacobian matrix in the hypergradient of BO, and hence fully eliminates all second-order computations. We call our algorithm as ESJ (which stands for the ES-based Jacobian method) and further extend it to the stochastic setting as ESJ-S. Theoretically, we characterize the convergence guarantee and computational complexity for our algorithms. Experimentally, we demonstrate the superiority of our proposed algorithms compared to the state of the art methods on various bilevel problems. Particularly, in our experiment in the few-shot meta-learning problem, we meta-learn the twelve millions parameters of a ResNet-12 network over the miniImageNet dataset, which evidently demonstrates the scalability of our ES-based bilevel approach and its feasibility in the large-scale setting.

## 1 Introduction

Bilevel optimization has recently arisen as a powerful tool to capture various modern machine learning problems, including meta-learning (Bertinetto et al., 2018; Franceschi et al., 2018; Rajeswaran et al., 2019; Ji et al., 2020a; Liu et al., 2021a), hyperparamater optimization (Franceschi et al., 2018; Shaban et al., 2019), neural architecture search (Liu et al., 2018a; Zhang et al., 2021), etc. Bilevel optimization generally takes the following mathematical form:

$$\min_{x \in \mathbb{R}^p} \Phi(x) := f(x, y^*(x)) \quad \text{s.t.} \quad y^*(x) = \arg\min_{y \in \mathbb{R}^d} g(x, y), \tag{1}$$

where the outer and inner objectives $f : \mathbb{R}^p \times \mathbb{R}^d \to \mathbb{R}$ and $g : \mathbb{R}^p \times \mathbb{R}^d \to \mathbb{R}$ are both continuously differentiable with respect to (w.r.t.) the inner and outer variables $x \in \mathbb{R}^p$ and $y \in \mathbb{R}^d$. In many machine learning problems, the loss functions $f$ and $g$ take a finite-sum form over a set of given data $\mathcal{D}_{n,m} = \{\xi_i, \zeta_j, i = 1, ..., n, j = 1, ..., m\}$ as follows:

$$f(x, y) = \frac{1}{n} \sum_{i=1}^n F(x, y; \xi_i), \quad g(x, y) = \frac{1}{m} \sum_{i=1}^m G(x, y; \zeta_i) \tag{2}$$

where the sample sizes $n$ and $m$ are typically very large. In this work, we consider a popular setting where the total objective function $\Phi(x)$ in eq. (1) is *possibly nonconvex* w.r.t. $x$ and the inner function $g$ is *strongly convex* w.r.t. $y$.

Gradient-based methods have served as a popular tool for solving bilevel optimization problems. Two types of approaches have been widely used: the iterative differentiation (ITD) method (Domke, 2012; Maclaurin et al., 2015; Finn et al., 2017; Franceschi et al., 2017; Shaban et al., 2019; Rajeswaran et al., 2019; Liu et al., 2020a) and the approximate iterative differentiation (AID) method (Domke, 2012; Pedregosa, 2016; Gould et al., 2016; Liao et al., 2018; Lorraine et al., 2020). In the ITD method, the objective value $f(x, \hat{y}(x))$ is first computed using an approximation point $\hat{y}(x)$ close to $y^*(x)$, and the hypergradient $\nabla \Phi(x) = \frac{\partial f(x, y^*(x))}{\partial x}$ is then approximated by $\frac{\partial f(x, \hat{y}(x))}{\partial x}$ using either reverse or forward mode automatic differentiation. Alternatively, the AID method leverages

the Implicit Function Theorem (IFT) to establish an implicit expression for the hypergradient. Due to the bilevel structure of the problem, these gradient-based ITD and AID approaches typically involve second-order matrix computations. Existing efficient implementations of these methods often adopt Jacobian- or/and Hessian-vector computations, which can still be very costly in practice for high-dimensional problems with deep neural networks.

To overcome the computational challenge of current gradient-based methods, Evolution Strategies (ES) can be a good candidate to eliminate the second-order computations, which uses function values for estimating its gradient in blackbox optimization (Nesterov & Spokoiny, 2017). An ES-based bilevel optimizer has recently been proposed in Gu et al. (2021) for hyperparameter optimization, which uses the ES method to approximate the hypergradient of $\Phi(x)$, treating $\Phi(x)$ fully as a blackbox objective. However, the **hypergradient** in bilevel optimization has a much more involved structure (see eq. (3)) than the **gradient** in standard minimization problems, and hence direct use of the ES method without exploiting the structure of the hypergradient can encounter a large estimation bias and result in poor performance in practice, especially in deep learning as demonstrated in our experiments in Section 4. This hence motivates the following intriguing question:

*Can we design a better hypergradient estimator based on the ES method by leveraging the analytical structure of the hypergradient, and then use such an estimator to construct more efficient ES-based bilevel optimizers, which are scalable for high-dimensional problems?*

Furthermore, since the hypergradient in bilevel optimization is much more challenging to estimate than the gradient in standard minimization problems, it is unclear whether the ES-based estimator can yield a provably convergent algorithm for bilevel optimization. So far the ES-based bilevel optimizer in Gu et al. (2021) did not come with a polynomial-time complexity guarantee. Our empirical experiments in Section 4 show that such an algorithm fail to converge even with shallow neural networks. This thus motivates the second question we ask:

*Can the ES method lead to bilevel optimizers with theoretical finite-time convergence guarantee?*

This paper provides the affirmative answers to both questions.

## 1.1 MAIN CONTRIBUTIONS

**Novel ES-based Jacobian bilevel optimizer.** We propose a novel bilevel optimizer with the **ES**-based **J**acobian estimation, which we call as ESJ. In contrast to the existing ES-based optimizer in Gu et al. (2021), which estimates the hypergradient by treating the outer-level objective $\Phi(x)$ fully as a blackbox, ESJ estimates only the *respone* Jacobian matrix (i.e., gradient of $y^*(x)$ with respect to $x$), which is the major computational bottleneck, and then leverages the analytical structure of the hypergradient to construct a much more accurate estimator. Further, ESJ has only gradient computations, and is computationally much more efficient than existing AID and ITD bilevel optimizers that require Hessian- and/or Jacobian-vector product computations. We further propose a stochastic bilevel optimizer ESJ-S to allow the algorithm scalable over large dataset.

**Convergence guarantee.** Theoretically, we characterize the convergence rate and their computational complexity for both ESJ and ESJ-S to achieve an $\epsilon$-accurate solution. Technically, in contrast to the standard analysis of ES-based methods on the smoothed blackbox function values, we develop tools to analyze the ES estimator on the output of the execution of an inner optimizer. Such an analysis can be of independent interest for ES-based bilevel optimization.

**Superior performance.** Experimentally, our algorithms ESJ and ESJ-S achieve superior performance compared to the current state of the art bilevel optimizers. Our approach is not only efficient and scalable, but also stable and robust across various bilevel problems. In particular, on the few-shot meta-learning problem, we meta-learn the twelve millions weights of a ResNet-12 network. Previous applications of other bilevel algorithms to meta-learning were limited only to settings where the backbone network has only up to few hundreds of thousands of parameters.

## 1.2 RELATED WORK

**Bilevel optimization.** Bilevel optimization has been studied for decades since Bracken & McGill (1973). A variety of bilevel optimization algorithms were then developed, including constraint-based approaches (Hansen et al., 1992; Shi et al., 2005; Moore, 2010), approximate implicit differ-

entiation (AID) approaches (Domke, 2012; Pedregosa, 2016; Gould et al., 2016; Liao et al., 2018; Lorraine et al., 2020) and iterative differentiation (ITD) approaches (Domke, 2012; Maclaurin et al., 2015; Franceschi et al., 2017; Finn et al., 2017; Shaban et al., 2019; Rajeswaran et al., 2019; Liu et al., 2020a). These methods often suffer from expensive computations of second-order information (e.g., Hessian-vector products). Hessian-free algorithms were recently proposed based on interior-point method (Liu et al., 2021b) and Evolution Strategies (ES) (Gu et al., 2021). This paper proposes a more efficient ES-based approach via exploiting the benign structure of the hypergradient.

Recently, the convergence rate has been established for gradient-based (Hessian involved) bilevel algorithms (Grazzi et al., 2020; Ji et al., 2021; Rajeswaran et al., 2019; Ji & Liang, 2021). This paper provides the convergence analysis for the proposed Hessian-free ES-based approach.

**Stochastic bilevel optimization.** Ghadimi & Wang (2018); Ji et al. (2021); Hong et al. (2020) proposed stochastic gradient descent (SGD) type bilevel optimization algorithms by employing Neumann series for Hessian-inverse-vector approximation. Recent works (Khanduri et al., 2021a;b; Chen et al., 2021; Guo et al., 2021; Guo & Yang, 2021; Yang et al., 2021) then leveraged momentum-based variance reduction to further reduce the computational complexity of existing SGD-type bilevel optimizers. In this paper, we propose a stochastic ES-based method, which eliminate the computation of second-order information required by all aforementioned stochastic methods.

Please see Appendix B for more works about applications of bilevel optimization and ES methods.

## 2 PROPOSED ALGORITHMS

We describe our proposed *deterministic* and *stochastic* algorithms for bilevel optimization.

### 2.1 HYPERGRADIENTS

The key step in popular gradient-based bilevel optimizers is the estimation of the hypergradient (i.e., the gradient of the objective with respect to the outer variable $x$), which takes the following form:

$$\nabla \Phi(x) = \nabla_x f(x, y^*(x)) + \mathcal{J}_*(x)^\top \nabla_y f(x, y^*(x)) \tag{3}$$

where the Jacobian matrix $\mathcal{J}_*(x) = \frac{\partial y^*(x)}{\partial x} \in \mathbb{R}^{d \times p}$. Following Lorraine et al. (2020), it can be seen that $\nabla \Phi(x)$ contains two components: the **direct** gradient $\nabla_x f(x, y^*(x))$ and the **indirect** gradient $\mathcal{J}_*(x)^\top \nabla_y f(x, y^*(x))$. The direct component can be efficiently computed using the existing automatic differentiation techniques. The indirect component, however, is computationally much more complex, because $\mathcal{J}_*(x)$ takes the form of $\mathcal{J}_*(x) = - \left[ \nabla_y^2 g(x, y^*(x)) \right]^{-1} \nabla_x \nabla_y g(x, y^*(x))$ (if $\nabla_y^2 g(x, y^*(x))$ is invertible), which contains the Hessian inverse and the second-order mixed derivative. Some approaches mitigate the issue by designing Jacobian-vector and Hessian-vector products (Pedregosa, 2016; Franceschi et al., 2017; Grazzi et al., 2020) to replace second-order computations. But these algorithms still have poor scalability to modern large-scale bilevel problems with high-dimensional variables such as neural network parameters. We next introduce the ES approach which is at the core of our idea for designing scalable Hessian-free bilevel optimizers.

### 2.2 EVOLUTION STRATEGIES (ES) METHOD

Evolution Strategies (ES) is a powerful technique to estimate the gradient of a function based on function values, when it is not feasible (such as in black-box problems) or computationally costly to evaluate the gradient. The idea of the ES method in Nesterov & Spokoiny (2017) is to approximate the gradient of a general black-box function $h : \mathbb{R}^n \to \mathbb{R}$ using the following oracle based only on the function values (i.e., the zeroth-order information)

$$\widehat{\nabla} h(x; u) = \frac{h(x + \mu u) - h(x)}{\mu} u \tag{4}$$

where $u \in \mathbb{R}^n$ is a Gaussian random vector and $\mu > 0$ is the smoothing parameter. Such an oracle can be shown to be an unbiased estimator of the gradient of the smoothed function $\mathbb{E}_u [h(x + \mu u)]$.

### 2.3 ES-BASED JACOBIAN AND ITS ENABLED BILEVEL OPTIMIZERS

**Drawback of an existing ES-based bilevel optimizer.** In Gu et al. (2021), the ES method was directly applied to estimate the hypergradient of $\Phi(x)$, treating $\Phi(x)$ as a blackbox objective. Since

the hypergradient has a complex form as in eq. (3) and can be very sensitive to both inner and outer functions, such an estimation will likely have a large bias error. As our experiments in Section 4 demonstrate, such a method is not robust and even fails to converge in complex bilevel optimization problems (such as with neural network parameters).

Our key idea is to exploit the analytical structure of the hypergradient in eq. (3), where the derivatives $\nabla_x f(x, y^*(x))$ and $\nabla_y f(x, y^*(x))$ can be computed efficiently and accurately, and then use the ES method only to estimate the Jacobian $\mathcal{J}_*(x)$, which is the major term posing computational difficulty. In this way, our estimation of the hypergradient can be much more accurate and reliable.

We propose the following novel ES-based Jacobian estimator, which contains two ingredients: **(i)** for a given $x$, apply an algorithm to solve the inner optimization problem and use the output as an approximation of $y^*(x)$; for example, the output $y^N(x)$ of $N$ gradient descent steps of the inner problem can serve as an estimate for $y^*(x)$. Then $\mathcal{J}_N(x) = \frac{\partial y^N(x)}{\partial x}$ serves as an estimate of $\mathcal{J}_*(x)$; and **(ii)** construct an ES-based Jacobian estimator $\hat{\mathcal{J}}_N(x; u) \in R^{d \times p}$ for $\mathcal{J}_N(x)$ as

$$\hat{\mathcal{J}}_N(x; u) = \frac{y^N(x + \mu u) - y^N(x)}{\mu} u^\top \tag{5}$$

where $u \in \mathbb{R}^p$ is a Gaussian vector with independent and identically distributed (i.i.d.) entries. Then for any vector $v \in \mathbb{R}^d$, the Jacobian-vector product can be efficiently computed using only vector-vector dot product $\hat{\mathcal{J}}_N(x; u)^\top v = \langle \delta(x; u), v \rangle u$, where $\delta(x; u) = \frac{y^N(x + \mu u) - y^N(x)}{\mu} \in \mathbb{R}^d$.

**ESJ: A novel bilevel optimizer with ES Jacobian oracles.** We design a bilevel optimizer (see Algorithm 1) using the ES-based Jacobian oracle in eq. (5), which we call as the **ESJ** algorithm. At each step $k$ of the algorithm, ESJ runs an $N$-step full GD to approximate $y_k^N(x_k)$. ESJ then samples $Q$ Gaussian vectors $\{u_{k,j} \in \mathcal{N}(0, I), j = 1, ..., Q\}$, and for each sample $u_{k,j}$, runs an $N$-step full GD to approximate $y_k^N(x_k + \mu u_{k,j})$, and then computes the Jacobian estimator $\hat{\mathcal{J}}_N(x; u_{k,j})$ as in eq. (5). Then the sample average over the $Q$ estimators is used for constructing the following hypergradient estimator for updating the outer variable $x$.

$$\begin{aligned} \widehat{\nabla} \Phi(x_k) &= \nabla_x f(x_k, y_k^N) + \frac{1}{Q} \sum_{j=1}^Q \hat{\mathcal{J}}_N^T(x_k; u_{k,j}) \nabla_y f(x_k, y_k^N) \\ &= \nabla_x f(x_k, y_k^N) + \frac{1}{Q} \sum_{j=1}^Q \langle \delta(x_k; u_{k,j}), \nabla_y f(x_k, y_k^N) \rangle u_{k,j}. \end{aligned} \tag{6}$$

Computationally, in contrast to the existing AID and ITD based bilevel optimizers (Pedregosa (2016), Franceschi et al. (2018), Grazzi et al. (2020)) that contains the complex Hessian- and/or Jacobian-vector product computations, ESJ has only gradient computations, and hence is computationally much more efficient as demonstrated in our experiments.

**ESJ-S: A stochastic bilevel optimizer with ES Jacobian oracles.** For the finite-sum problem with the objective functions given in eq. (2), we design a **stochastic** bilevel optimizer (see Algorithm 2 in Appendix A) based on ES Jacobian oracles, which we call as **ESJ-S**.

Differently from Algorithm 1, which applies GD updates to find $y^N(x_k)$, ESJ-S adopts $N$ stochastic gradient descent (SGD) steps to find $\{Y_k^N, Y_{k,1}^N, ..., Y_{k,Q}^N\}$ to the inner problem, each with the outer variable set to be $x_k + \mu u_{k,j}$. Note that all SGD runs follow the same batch sampling path $\{\mathcal{S}_0, ..., \mathcal{S}_{N-1}\}$. The Jacobian estimator $\hat{\mathcal{J}}_N(x_k; u_{k,j})$ can then be computed as in eq. (5). At the outer level, ESJ-S samples a new batch $\mathcal{D}_F$ independently from the inner batches $\{\mathcal{S}_0, ..., \mathcal{S}_{N-1}\}$ to evaluate the stochastic gradients $\nabla_x F(x_k, Y_k^N; \mathcal{D}_F)$ and $\nabla_y F(x_k, Y_k^N; \mathcal{D}_F)$. The hypergradient is then estimated as follows.

$$\widehat{\nabla} \Phi(x_k) = \nabla_x F(x_k, Y_k^N; \mathcal{D}_F) + \frac{1}{Q} \sum_{j=1}^Q \langle \delta(x_k; u_{k,j}), \nabla_y F(x_k, Y_k^N; \mathcal{D}_F) \rangle u_{k,j}. \tag{7}$$

## 3 CONVERGENCE ANALYSIS

### 3.1 TECHNICAL ASSUMPTIONS

In this work, we focus on the following types of loss functions.

**Assumption 1.** *The inner function $g(x, y)$ is $\mu_g$-strongly convex with respect to the variable $y$. For the finite-sum case, the same assumption holds for $\nabla G(x, y; \zeta)$.*

---

**Algorithm 1** Bilevel optimizer via ES Jacobians (ESJ)

---

1: **Input:** lower- and upper-level stepsizes $\alpha, \beta > 0$, initializations $x_0 \in \mathbb{R}^p$ and $y_0 \in \mathbb{R}^d$, inner and outer iterations numbers $K$ and $N$, and number of Gaussian vectors $Q$.
2: **for** $k = 0, 1, 2, ..., K$ **do**
3:    Set $y_k^0 = y_0, \quad y_{k,j}^0 = y_0, j = 1, ..., Q$
4:    **for** $t = 1, 2, ..., N$ **do**
5:       Update $y_k^t = y_k^{t-1} - \alpha \nabla_y g(x_k, y_k^{t-1})$
6:    **end for**
7:    **for** $j = 1, ..., Q$ **do**
8:       Generate $u_{k,j} = \mathcal{N}(0, I) \in \mathbb{R}^p$
9:       **for** $t = 1, 2, ..., N$ **do**
10:          Update $y_{k,j}^t = y_{k,j}^{t-1} - \alpha \nabla_y g\left(x_k + \mu u_{k,j}, y_{k,j}^{t-1}\right)$
11:       **end for**
12:       Compute $\delta_j = \frac{y_{k,j}^N - y_k^N}{\mu}$
13:    **end for**
14:    Compute $\widehat{\nabla}\Phi(x_k) = \nabla_x f(x_k, y_k^N) + \frac{1}{Q}\sum_{j=1}^{Q} \left\langle \delta_j, \nabla_y f(x_k, y_k^N) \right\rangle u_{k,j}$
15:    Update $x_{k+1} = x_k - \beta \widehat{\nabla}\Phi(x_k)$
16: **end for**

---

We take the following assumptions on the inner and outer loss functions $g(x, y)$ and $f(x, y)$, as also adopted in Ghadimi & Wang (2018); Ji et al. (2021); Yang et al. (2021).

**Assumption 2.** *Let $w = (x, y) \in \mathbb{R}^{d+p}$. The gradient $\nabla g(w)$ is $L_g$-Lipschitz continuous, i.e., for any $w_1, w_2 \in \mathbb{R}^{d+p}$, $\left\| \nabla g(w_1) - \nabla g(w_2) \right\| \leq L_g \left\| w_1 - w_2 \right\|$; further, the derivatives $\nabla_y^2 g(w)$ and $\nabla_x \nabla_y g(w)$ are $\rho$- and $\tau$-Lipschitz continuous, i.e, $\left\| \nabla_y^2 g(w_1) - \nabla_y^2 g(w_2) \right\|_F \leq \rho \left\| w_1 - w_2 \right\|$ and $\left\| \nabla_x \nabla_y g(w_1) - \nabla_x \nabla_y g(w_2) \right\|_F \leq \tau \left\| w_1 - w_2 \right\|$. For the finite-sum case, the same assumptions hold for $G(w; \zeta)$.*

**Assumption 3.** *Let $w = (x, y) \in \mathbb{R}^{d+p}$. The objective $f(w)$ and its gradient $\nabla f(w)$ are $M$- and $L_f$-Lipschitz continuous, i.e., for any $w_1, w_2 \in \mathbb{R}^{d+p}$,*

$$\left| f(w_1) - f(w_2) \right| \leq M \left\| w_1 - w_2 \right\|, \quad \left\| \nabla f(w_1) - \nabla f(w_2) \right\| \leq L_f \left\| w_1 - w_2 \right\|.$$

*For the finite-sum case, the same assumptions hold for $F(w; \xi)$.*

For the finite-sum setting, we take the standard bounded-variance assumption on $\nabla G(w; \zeta)$.

**Assumption 4.** *Gradient $\nabla G(w; \zeta)$ has a bounded variance, i.e., $\mathbb{E}_\zeta \| \nabla G(w; \zeta) - \nabla g(w) \|^2 \leq \sigma^2$ for some constant $\sigma \geq 0$.*

## 3.2   Convergence Analysis for ESJ

Different from the standard zeroth-order analysis for a blackbox function, here we develop new techniques to analyze the ES-based estimator that depend on the entire inner optimization trajectory, which is specific to bilevel optimization. We first establish the following essential proposition which characterizes the Lipshitzness property of the approximate Jacobian matrix $\mathcal{J}_N(x) = \frac{\partial y^N(x)}{\partial x}$.

**Proposition 1.** *Suppose that Assumptions 1 and 2 hold. Let $L_\mathcal{J} = \left(1 + \frac{L}{\mu_g}\right)\left(\frac{\tau}{\mu_g} + \frac{\rho L}{\mu_g^2}\right)$, with $L = \max\{L_f, L_g\}$. Then, the Jacobian $\mathcal{J}_N(x)$ is Lipschitz continuous with constant $L_\mathcal{J}$:*

$$\left\| \mathcal{J}_N(x_1) - \mathcal{J}_N(x_2) \right\|_F \leq L_\mathcal{J} \left\| x_1 - x_2 \right\| \qquad \forall x_1, x_2 \in \mathbb{R}^p. \tag{8}$$

We next provide an upper-bound on the hypergradient estimation error for ESJ.

**Proposition 2.** *Suppose that Assumptions 1, 2, and 3 hold. Consider the ESJ algorithm. Its hypergradient estimation error can be upper-bounded as:*

$$\mathbb{E}\left\| \widehat{\nabla}\Phi(x_k) - \nabla\Phi(x_k) \right\|^2 \leq \mathcal{O}\left( (1 - \alpha\mu_g)^N + \frac{p}{Q} + \mu^2 d p^3 + \frac{\mu^2 d p^4}{Q} \right) \tag{9}$$

*where $\mathbb{E}[\cdot]$ is conditioned on $x_k$ and taken over the Gaussian vectors $\{u_{k,j} : j = 1, ..., Q\}$.*

By using the smoothness property in Proposition 1 and the upper-bound in Proposition 2, we provide the following characterization of the convergence rate for ESJ.

**Theorem 1** (Convergence of ESJ). *Suppose that Assumptions 1, 2, and 3 hold. Choose the inner- and outer-loop stepsizes respectively as $\alpha \leq \frac{1}{L}$ and $\beta = \frac{1}{4L_\Phi}$, where $L_\Phi = L + \frac{2L^2 + \tau M^2}{\mu_g} + \frac{\rho LM + L^3 + \tau ML}{\mu_g^2} + \frac{\rho L^2 M}{\mu_g^3}$. Then, the iterates $x_k$ for $k = 0, ..., K-1$ of ESJ in Algorithm 1 satisfy:*

$$\frac{1}{K} \sum_{k=0}^{K-1} \mathbb{E} \big\| \nabla \Phi(x_k) \big\|^2 \leq \frac{16L_\phi (\Phi(x_0) - \Phi^*)}{K} + \Delta_1 \tag{10}$$

*with $\Phi^* = \inf_x \Phi(x)$ and $\Delta_1 = \mathcal{O}\left( (1-\alpha\mu_g)^N + \frac{p}{Q} + \frac{\mu^2 dp^4}{Q} + \mu^2 dp^3 \right)$.*

*Further, choose $K$, $N$, $\mu$, and $Q$ at the order of $\mathcal{O}(\frac{1}{\epsilon})$, $\mathcal{O}(\log \frac{1}{\epsilon})$, $\mathcal{O}(\min\{\frac{1}{p}\sqrt{\frac{\epsilon}{dp}}, \frac{1}{p}\sqrt{\frac{1}{dp}}\})$ and $\mathcal{O}(\frac{p}{\epsilon})$. Then ESJ achieves an $\epsilon$-accurate stationary point with $\mathcal{O}(\frac{p}{\epsilon^2} \log \frac{1}{\epsilon})$ computations of the gradient $\nabla_y g(x, y)$ and $\mathcal{O}(\frac{1}{\epsilon})$ computations of the gradients $\nabla_x f(x, y)$ and $\nabla_y f(x, y)$.*

Theorem 1 characterizes the sublinear convergence of ESJ with respect to the number $K$ of outer iterations due to the nonconvexity of the outer objective. The convergence error in $\Delta_1$ is due to the estimation error of the Jacobian matrix $\mathcal{J}_*$ via our ES oracle, which captures three types of errors: (a) the approximation error between $\mathcal{J}_N$ and $\mathcal{J}_*$ via inner-loop gradient descent, which decreases exponentially w.r.t. the number $N$ of inner iterations due to the strong convexity of the inner objective; (b) the error between our ES oracle and the Jacobian $\mathcal{J}_\mu$ of the smoothed output $\mathbb{E}_u y^N (x_k + \mu u)$, which decreases sublinearly w.r.t. the batch size $Q$, and (c) the error between the Jacobians $\mathcal{J}_N$ and $\mathcal{J}_\mu$, which can be controlled by the smoothness parameter $\mu$.

Theorem 1 also indicates that ESJ requires only gradient computations to converge, and eliminates Hessian- and Jacobian-vector products required by the existing AID and ITD based bilevel optimizers (Pedregosa (2016), Franceschi et al. (2018), Grazzi et al. (2020)). Thus, ESJ is computationally much more efficient particularly for high-dimensional problems.

### 3.3 Convergence Analysis for ESJ-S

In this section, we apply the stochastic algorithm ESJ-S to the finite-sum objective in eq. (2) and analyze its convergence rate. We first note that the Lipschitzness property established in Proposition 1 is general and can also be used here. The following proposition establishes an upper bound on the estimation error of Jacobian $\mathcal{J}_*$ by $\mathcal{J}_N = \frac{\partial Y^N}{\partial x}$, where $Y^N$ is the output of $N$ inner SGD updates.

**Proposition 3.** *Suppose that Assumptions 1, 2, and 4 hold. Consider the ESJ-S algorithm. Choose the inner-loop stepsize as $\alpha = \frac{2}{L + \mu_g}$, where $L = \max\{L_f, L_g\}$. Then, we have:*

$$\mathbb{E}\big\| \mathcal{J}_N - \mathcal{J}_* \big\|_F^2 \leq C_\gamma^N \frac{L^2}{\mu_g^2} + \frac{\lambda (L+\mu_g)^2 (1-\alpha\mu_g) C_\gamma^{N-1}}{(L+\mu_g)^2 (1-\alpha\mu_g) - (L-\mu_g)^2} + \frac{\Gamma}{1-C_\gamma}.$$

*where $\lambda$, $\Gamma$, and $C_\gamma < 1$ are constants (see appendix G for their forms).*

We next provide an upper-bound on the estimation error of the hypergradient by $\widehat{\nabla}\Phi(x_k)$ in eq. (7).

**Proposition 4.** *Suppose that Assumptions 1, 2, 3, and 4 hold. Consider the ESJ-S algorithm. Set the inner-loop stepsize as $\alpha = \frac{2}{L+\mu_g}$ where $L = \max\{L_f, L_g\}$. Then, we have:*

$$\mathbb{E}\big\| \widehat{\nabla}\Phi(x_k) - \nabla\Phi(x_k) \big\|^2 \leq \mathcal{O}\left( (1-\alpha\mu_g)^N + \frac{1}{S} + \frac{1}{D_f} + \frac{p}{Q} + \mu^2 dp^3 + \frac{\mu^2 dp^4}{Q} \right)$$

*where $S$ and $D_f$ are the sizes of the inner and outer mini-batches, respectively.*

Based on Propositions 1, 3, and 4, we characterize the convergence rate for ESJ-S.

**Theorem 2** (Convergence of ESJ-S). *Suppose that Assumptions 1, 2, 3, and 4 hold. Set the inner- and outer-loop stepsizes respectively as $\alpha = \frac{2}{L+\mu_g}$ and $\beta = \frac{1}{4L_\Phi}$, where $L = \max\{L_f, L_g\}$ and the constant $L_\Phi$ are defined in Theorem 1. Then, the iterates $x_k, k = 0, ..., K-1$ of ESJ-S satisfy:*

$$\frac{1}{K} \sum_{k=0}^{K-1} \mathbb{E}\big\| \nabla\Phi(x_k) \big\|^2 \leq \frac{16(\Phi(x_0) - \Phi^*) L_\Phi}{K} + \Delta_2, \tag{11}$$

*where $\Delta_2 = \mathcal{O}\left((1 - \alpha\mu_g)^N + \frac{1}{S} + \frac{1}{D_f} + \frac{p}{Q} + \frac{\mu^2 dp^4}{Q} + \mu^2 dp^3\right)$.*

*For ESJ-S to attain an $\epsilon$-accurate stationary point, it suffices to select $K$, $S$, and $D_f$ at the order of $\mathcal{O}(\frac{1}{\epsilon})$, and $N$, $\mu$, and $Q$ respectively at the order of $\mathcal{O}(\log \frac{1}{\epsilon})$, $\mathcal{O}(\min\{\frac{1}{p}\sqrt{\frac{\epsilon}{dp}}, \frac{1}{p}\sqrt{\frac{1}{dp}}\})$ and $\mathcal{O}(\frac{p}{\epsilon})$, which thus amount to $\mathcal{O}(\frac{p}{\epsilon^3} \log \frac{1}{\epsilon})$ computations of the gradient $\nabla_y G(x, y, \xi)$ and $\mathcal{O}(\frac{1}{\epsilon^2})$ computations of the gradients $\nabla_x F(x, y, \xi)$ and $\nabla_y F(x, y, \xi)$.*

Comparing to the convergence error $\Delta_1$ in Theorem 1 for the *deterministic* algorithm ESJ, Theorem 2 for the *stochastic* algorithm ESJ-S captures two more sublinearly decreasing error terms $\frac{1}{S}$ and $\frac{1}{D_f}$ respectively due to the sampling of inner and outer batches to estimate the objectives.

To compare between ESJ-S and ESJ on the finite-sum objective in eq. (2), Theorem 1 suggests that ESJ attains an $\epsilon$-accurate stationary point of eq. (2) with $\mathcal{O}(\frac{pm}{\epsilon^2} \log \frac{1}{\epsilon})$ computations of $\nabla G$ and $\mathcal{O}(\frac{n}{\epsilon})$ computations of $\nabla F$. As a comparison, Theorem 2 suggests that ESJ-S achieves $\mathcal{O}(\frac{p}{\epsilon^3} \log \frac{1}{\epsilon})$ computations of $\nabla G$ and $\mathcal{O}(\frac{1}{\epsilon^2})$ computations of $\nabla F$, which outperforms ESJ in the large sample regime where the sample sizes $n, m > \frac{1}{\epsilon}$. This also reflects the advantage of SGD over GD.

## 4 EXPERIMENTS

We validate our algorithms over three bilevel problems: **shallow hyper-representation (HR)** with linear/2-layer net embedding model on synthetic data, **deep HR with LeNet network** (LeCun et al., 1998) on MNIST dataset, and **few-shot meta-learning** with **ResNet-12** on miniImageNet dataset. We run all models using a single NVIDIA Tesla P100 GPU. Hyperparamter Optimization (HO) experiments can be found in Appendix D. All running time are in seconds.

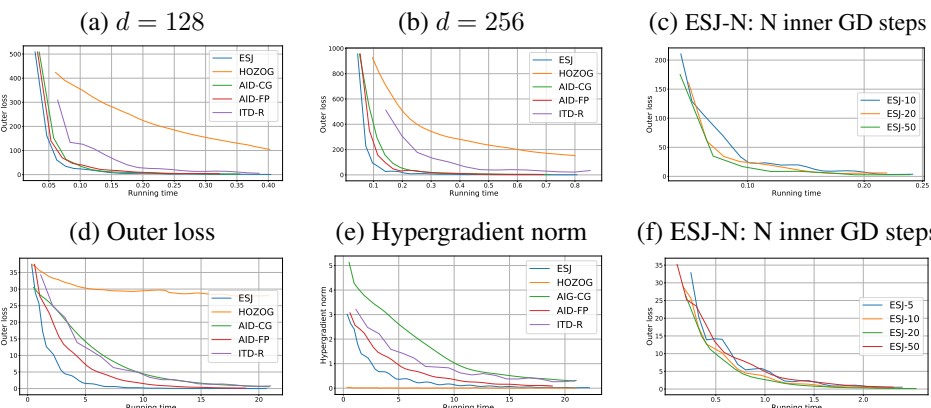

Figure 1: First row: HR with linear embedding model. Second row: HR with two-layer net.

### 4.1 SHALLOW HYPER-REPRESENTATION ON SYNTHETIC DATA

The hyper-representation (HR) problem (Franceschi et al., 2018; Grazzi et al., 2020) searches for a regression (or classification) model following a two-phased optimization process. The inner-level identifies the optimal linear regressor parameters $w$, and the outer level solves for the optimal embedding model (i.e., representation) parameters $\lambda$. Mathematically, the problem can be modeled by the following bilevel optimization:

$$\min_{\lambda \in \mathbb{R}^p} f(\lambda) = \frac{1}{2n_1} \|T(X_1; \lambda)w^*(\lambda) - Y_1\|^2 \text{ s.t. } w^*(\lambda) = \underset{w \in \mathbb{R}^d}{\operatorname{argmin}} \frac{1}{2n_2} \|T(X_2; \lambda)w - Y_2\|^2 + \frac{\gamma}{2}\|w\|^2 \quad (12)$$

where $X_2 \in \mathbb{R}^{n_2 \times m}$ and $X_1 \in \mathbb{R}^{n_1 \times m}$ are matrices of synthesized training and validation data, and $Y_2 \in \mathbb{R}^{n_2}$, $Y_1 \in \mathbb{R}^{n_1}$ are the corresponding response vectors. In the case of shallow HR, the embedding function $T(\cdot; \lambda)$ is either a linear transformation or a two-layer network. We generate data matrices $X_1, X_2$ and labels $Y_1, Y_1$ following the same process in Grazzi et al. (2020).

We compare our ESJ algorithm with the baseline bilevel optimizers AID-FP, AID-CG, ITD-R, and HOZOG (see Appendix C.1 for details about the baseline algorithms and hyperparameters used).

Figure 1 show the performance comparison among the algorithms under linear and two-layer net embedding models. It can be observed that for both cases, our proposed method ESJ converges faster than all the other approaches, and the advantage of ESJ becomes more significant in Figure 1 (d), which is under a higher-dimensional model of a two-layer net. In particular, ESJ outperforms the existing ES-based algorithm HOZOG. This is because HOZOG uses the ES technique to approximate the entire hypergradient, which likely incurs a large estimation error. In contrast, our ESJ exploits the structure of the hypergradient and uses ES only to estimate the response Jacobian so that the estimation of hypergradient is more accurate. Such an advantage is more evident under a two-layer net model, where HOZOG does not converge as shown in Figure 1 (d). This can be explained by the flat hypergradient norm as shown in Figure 1 (e), which indicates that the hypergradient estimator in HOZOG fails to provide a good descent direction for the outer optimizer. Figure 1 (c) and (f) further show that the convergence of ESJ does not change substantially with the number $N$ of inner GD steps, and hence tuning of $N$ in practice is not costly.

## 4.2 DEEP HYPER-REPRESENTATION ON MNIST DATASET

In order to demonstrate the advantage of our proposed algorithms in large neural net models, we perform deep hyper-representation to classify MNIST images by learning an entire LeNet network. The corresponding bilevel problem is given by

$$
\begin{aligned}
\min_{\lambda} & \ \mathcal{L}_{\text{out}}(\lambda) := \frac{1}{|\mathcal{D}_{\text{out}}|} \sum_{(x_i,y_i)\in\mathcal{D}_{\text{out}}} \mathcal{L}\left(w^*(\lambda)f(x_i;\lambda), y_i\right) \\
\text{s.t.} & \ \ w^*(\lambda) = \arg\min_{w\in\mathbb{R}^{c\times p}} \mathcal{L}_{\text{in}}(w,\lambda) := \frac{1}{|\mathcal{D}_{\text{in}}|} \sum_{(x_i,y_i)\in\mathcal{D}_{\text{in}}} \left(\mathcal{L}(wf(x_i;\lambda),y_i) + \frac{\beta}{2}\|w\|^2\right).
\end{aligned}
\tag{13}
$$

where $f(x_i;\lambda) \in \mathbb{R}^p$ corresponds to features extracted from data point $x_i$, $\mathcal{L}(\cdot,\cdot)$ is the cross-entropy loss function, $c = 10$ is the number of categories, and $\mathcal{D}_{\text{in}}$ and $\mathcal{D}_{\text{out}}$ are data used to compute respectively inner and outer loss functions. Since the sizes of $\mathcal{D}_{\text{out}}$ and $\mathcal{D}_{\text{in}}$ are large in the case of MNIST dataset, we apply the more efficient stochastic algorithm ESJ-S in Algorithm 2 with a minibatch size $B = 256$ to estimate the inner and outer losses $\mathcal{L}_{\text{in}}$ and $\mathcal{L}_{\text{out}}$.

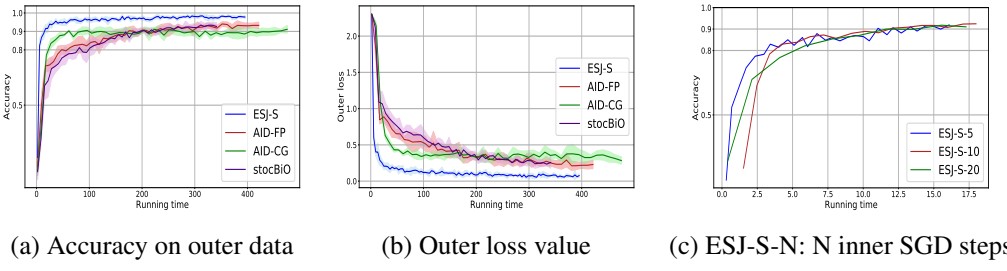

(a) Accuracy on outer data  (b) Outer loss value  (c) ESJ-S-N: N inner SGD steps

Figure 2: Deep HR on the MNIST dataset.

Figure 2 compares the classification accuracy on the outer dataset $\mathcal{D}_{\text{out}}$ among the different methods. Our algorithm ESJ-S converges with the fastest rate and attains the best accuracy with the lowest variance among all algorithms. Note that ESJ-S is able to attain the same accuracy of $0.98+$ obtained by the standard training of all parameters with one-phased optimization on the MNIST dataset using the same backbone network. All other methods fail to recover such a level of performance, and instead saturate around an accuracy of $0.93$. Further, Figure 2(c) indicates that the convergence of ESJ-S does not change substantially with the number $N$ of inner SGD steps. This demonstrates the robustness of our method when applied to complex function geometries such as deep nets.

## 4.3 FEW-SHOT META-LEARNING OVER MINIIMAGENET

To study our algorithms over larger neural nets, we study the few-shot image recognition problem, where classification tasks $\mathcal{T}_i, i = 1, ..., m$ are sampled over a distribution $\mathcal{P}_{\mathcal{T}}$. In particular, we consider the ANIL meta-learning method (Raghu et al., 2019; Ji et al., 2020a), where all tasks share common embedding features parameterized by $\phi$, and each task $\mathcal{T}_i$ has its task-specific parameter $w_i$ for $i = 1, ..., m$. More specifically, we set $\phi$ to be the parameters of the convolutional part of a deep CNN model (e.g., ResNet-12 network) and $w$ includes the parameters of the last classification layer. All model parameters $(\phi, w)$ are trained following a bilevel procedure. In the inner-loop, the base learner of each task $\mathcal{T}_i$ fixes $\phi$ and minimizes its loss function over a training set $\mathcal{S}_i$ to

obtain its adapted parameters $w_i^*$. At the outer stage, the meta-learner computes the test loss for each task $\mathcal{T}_i$ using the parameters $(\phi, w_i^*)$ over a test set $\mathcal{D}_i$, and optimizes the parameters $\phi$ of the common embedding function by minimizing the meta-objective $\mathcal{L}_{\text{meta}}$ over all classification tasks. The problem can be expressed as the following bilevel optimization

$$
\begin{aligned}
\min_{\phi}\ & \mathcal{L}_{\text{meta}}(\phi, \widetilde{w}^*) := \tfrac{1}{m} \sum_{i=1}^{m} \mathcal{L}_{\mathcal{D}_i}(\phi, w_i^*) \\
\text{s.t.}\quad & \widetilde{w}^* = \arg\min_{\widetilde{w}} \mathcal{L}_{\text{adapt}}(\phi, \widetilde{w}) := \tfrac{1}{m} \sum_{i=1}^{m} \mathcal{L}_{\mathcal{S}_i}(\phi, w_i),
\end{aligned}
\tag{14}
$$

where we collect all task-specific parameters into $\widetilde{w} = (w_1, ..., w_m)$ and the corresponding minimizers into $\widetilde{w}^* = (w_1^*, ..., w_m^*)$. The functions $\mathcal{L}_{\mathcal{S}_i}(\phi, w_i) = \frac{1}{|\mathcal{S}_i|} \sum_{\zeta \in \mathcal{S}_i} (\mathcal{L}(\phi, w_i; \zeta) + \mathcal{R}(w_i))$ and $\mathcal{L}_{\mathcal{D}_i}(\phi, w_i^*) = \frac{1}{|\mathcal{D}_i|} \sum_{\xi \in \mathcal{D}_i} \mathcal{L}(\phi, w_i^*; \xi)$ correspond respectively to the training and test loss functions for task $\mathcal{T}_i$, with $\mathcal{R}$ a strongly-convex regularizer and $\mathcal{L}$ a classification loss function. In our setting, since the task-specific parameters correspond to the weights of the last linear layer, the inner-level objective $\mathcal{L}_{\text{adapt}}(\phi, \widetilde{w})$ is strongly convex with respect to $\widetilde{w} = (w_1, ..., w_m)$. We note that the problem studied in Section 4.2 can be seen as single-task instances of the more general multi-task learning problem in eq. (14). However, in contrast to the problem in Section 4.2, the sizes of the datasets $\mathcal{D}_i$ and $\mathcal{S}_i$ are usually small in few-shot learning and full GD can be applied here. Hence, we use ESJ (Algorithm 1) here. Also since the number $m$ of tasks in few-shot classification datasets is often very large, it is preferable to sample a minibatch of i.i.d. tasks by $\mathcal{P}_{\mathcal{T}}$ at each meta (i.e., outer) iteration and update the meta parameters based on these tasks.

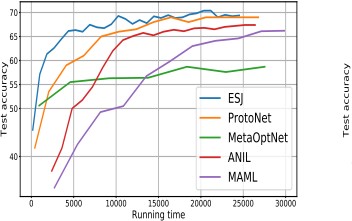
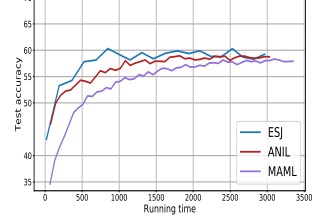

| Algorithm | Time (hours) |
|:---:|:---:|
| ESJ | 2.8 |
| MetaOptNet | 20+ |
| ProtNet | 4.4 |

(a) Test accuracy (ResNet-12)   (b) Test accuracy (CNN4)   (c) Time to reach 69% (ResNet-12)

Figure 3: 5way-5shot image classification on the miniImageNet dataset on single GPU.

We conduct few-shot meta-learning on the miniImageNet dataset (Vinyals et al., 2016) using two different backbone networks for feature extraction: ResNet-12 and CNN4 (Vinyals et al., 2016). The dataset description and hyperparameter details can be found in Appendix C.4. We compare our algorithm ESJ with four baseline methods for few-shot meta-learning **MAML** (Finn et al., 2017), **ANIL** (Raghu et al., 2019), **MetaOptNet** (Lee et al., 2019), and **ProtoNet** (Snell et al., 2017). We run their efficient Pytorch Lightning implementations available at the *learn2learn* repository (Arnold et al., 2019).

Figure 3(a) and (b) show that our algorithm ESJ converges faster than the other baseline methods. Also, Comparing Figure 3(a) and (b), the advantage of our method over the baselines **MAML** and **ANIL** becomes more significant as the size of the network increases. Further, Figure 3(c) shows that MetaOptNet did not reach 69% accuracy after 20 hours of training with ResNet-12 network. In comparison, our ESJ is able to attain 69% in less than three hours, which is about 1.5 times less than the time taken for ProtoNet to reach the same performance level. Both ESJ and ProtoNet saturate around 70% accuracy after 10 hours of training.

## 5   CONCLUSION

In this paper, we propose a novel ES-based approach for bilevel optimization, which eliminates all second-order computations in the gradient-based approaches. Compared to the existing ES-based algorithm, our approach explores the analytical structure of the hypergradient, and hence leads to much more efficient and accurate hypergradient estimation. Thus, our algorithm outperforms the existing algorithms in the experiments, particularly in the high-dimensional applications. We also characterize the convergence rate for our proposed algorithms and show that the polynomial-time complexity can be achieved. We anticipate that our approach will be useful for accelerating bilevel algorithms in various machine learning problems.

## 6 REPRODUCIBILITY CHECKLIST

To ensure reproducibility, we use the Machine Learning Reproducibility Checklist v2.0, Apr. 7 2020 (Pineau et al., 2021). An earlier version of this checklist (v1.2) was used for NeurIPS 2019 (Pineau et al., 2021).

- For all **models** and **algorithms** presented,
  - **A clear description of the mathematical settings, algorithm, and/or model.** We clearly describe all of the settings, formulations, and algorithms in Section 2.
  - **A clear explanation of any assumptions.** All assumptions are stated in Section 3.1 and details are clearly explained in Section 3.1.
  - **An analysis of the complexity (time, space, sample size) of any algorithm.** We provide the time/computational complexity analysis for our algorithms in Theorems 1 and 2.
- For any **theoretical claim**,
  - **A clear statement of the claim.** A clear statement of theoretical claims are made in Section 3.
  - **A complete proof of the claim.** The complete proofs of all claims are available in Appendix E, Appendix F, and Appendix G.
- For all **datasets** used, check if you include:
  - **The relevant statistics, such as number of examples.** We use widely adopted datasets MNIST and miniImageNet in Section 4. The related statistics can be seen at `http://yann.lecun.com/exdb/mnist/`.
  - **The details of train/validation/test splits.** We give this information in the Supplementary Appendix C.
  - **An explanation of any data that were excluded, and all pre-processing step.** We did not exclude any data or perform any pre-processing.
  - **For new data collected,a complete description of the data collection process, such as instructions to annotators and methods for quality control.** We do not collect or release new datasets.
- For all shared **code** related to this work, check if you include:
  - **Training code.** The training code is available in our code in supplementary material.
  - **Evaluation code.** The evaluation code is available in our code in supplementary material.
  - **(Pre-)trained model(s).** We do not release any pre-trained models.
- For all reported **experimental results**, check if you include:
  - **The range of hyper-parameters considered, method to select the best hyper-parameter configuration, and specification of all hyper-parameters used to generate results.** We provide all details of the hyper-parameter tuning in Supplementary Appendix C.
  - **A clear definition of the specific measure or statistics used to report results.** We use the classification accuracy on test-set and the loss on the train-set.
  - **A description of results with central tendency (e.g. mean) & variation (e.g. error bars).** We do not report the mean and standard deviation for experiments.
  - **The average runtime for each result, or estimated energy cost.** We report the running time of the algorithms in Section 4.
  - **A description of the computing infrastructure used.** All detailed descriptions are presented in Section 4.

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

# Supplementary Material

## A    STOCHASTIC BILEVEL OPTIMIZER BASED ON ES METHOD

In the following, we present the algorithm specification for our proposed stochastic bilevel optimizer based on ES method, which we call **ESJ-S**.

---

**Algorithm 2** Stochastic bilevel optimizer via ES Jacobians (ESJ-S)

---

1: **Input:** lower- and upper-level stepsizes $\alpha, \beta > 0$, initializations $x_0 \in \mathbb{R}^p$ and $y_0 \in \mathbb{R}^d$, inner and outer iterations numbers $K$ and $N$, and number of Gaussian vectors $Q$.
2: **for** $k = 0, 1, ..., K$ **do**
3:     Set $Y_k^0 = y_0$,    $Y_{k,j}^0 = y_0, j = 1, ..., Q$
4:     Generate $u_{k,j} = \mathcal{N}(0, I) \in \mathbb{R}^p$,    $j = 1, ..., Q$
5:     **for** $t = 1, 2, ..., N$ **do**
6:         Draw a sample batch $\mathcal{S}_{t-1}$
7:         Update $Y_k^t = Y_k^{t-1} - \alpha \nabla_y G(x_k, Y_k^{t-1}; \mathcal{S}_{t-1})$
8:         **for** $j = 1, ..., Q$ **do**
9:             Update $Y_{k,j}^t = Y_{k,j}^{t-1} - \alpha \nabla_y G\left(x_k + \mu u_{k,j}, Y_{k,j}^{t-1}; \mathcal{S}_{t-1}\right)$
10:        **end for**
11:    **end for**
12:    Compute $\delta_j = \frac{Y_{k,j}^N - Y_k^N}{\mu}$,    $j = 1, ..., Q$
13:    Draw a sample batch $\mathcal{D}_F$
14:    Compute $\widehat{\nabla}\Phi(x_k) = \nabla_x F(x_k, Y_k^N; \mathcal{D}_F) + \frac{1}{Q} \sum_{j=1}^Q \left\langle \delta_j, \nabla_y F(x_k, Y_k^N; \mathcal{D}_F) \right\rangle u_{k,j}$
15:    Update $x_{k+1} = x_k - \beta \widehat{\nabla}\Phi(x_k)$
16: **end for**

---

## B    ADDITIONAL RELATED WORK

In this section, we provide further related work on the applications of bilevel optimization and ES methods.

**Bilevel optimization applications.** Bilevel optimization has been employed in various applications such as few-shot meta-learning (Snell et al., 2017; Franceschi et al., 2018; Rajeswaran et al., 2019; Zügner & Günnemann, 2019; Ji et al., 2020a;b), hyperparameter optimization (Franceschi et al., 2017; Mackay et al., 2018; Shaban et al., 2019), neural architecture search (Liu et al., 2018a; Zhang et al., 2021), etc. This paper demonstrates the superior performance of the proposed ES-based bilevel optimizer in meta-learning and hyperparameter optimization.

**ES applications.** Evolution Strategies (ES) method has been studied for more than two decades (see review papers (Hansen, 2006; Varelas et al., 2018)). In particular, Nesterov & Spokoiny (2017) proposed a simple and effective ES-gradient via Gaussian smoothing, which was further extended to the stochastic setting by Ghadimi & Lan (2013). Such an ES technique has exhibited great effectiveness in various applications including meta-reinforcement learning (Song et al., 2019), hyperparameter optimization (Gu et al., 2021), adversarial machine learning (Ji et al., 2019; Liu et al., 2018b), min-imax optimization (Liu et al., 2020b; Xu et al., 2020), etc. This paper proposes a novel ES-based Jacobian estimator for accelerating bilevel optimization. The ES method has also been used for studying hyperparameter optimization problems. For example, Mackay et al. (2018) models the response function itself as a neural network (where each layer involves an affine transformation of hyperparameters) using the Self-Tuning Networks (STNs). An improved and more stable version of STNs was further proposed in Bae & Grosse (2020), which focused on accurately approximating the response Jacobian rather than the response function itself.

## C    Further Specifications for Experiments in Section 4

We note that the smoothing parameter $\mu$ (in Algorithms 1 and 2) was easy to set and a value of $0.1$ or $0.01$ yields a good starting point across all our experiments. The batch size $Q$ (in Algorithms 1 and 2) is fixed to $1$ (i.e., we use one Jacobian oracle) in all our experiments.

### C.1    Specifications on Baseline Bilevel Approaches in Section 4.1

We compare our algorithm ESJ with the following baseline methods:

- HOZOG (Gu et al., 2021): a hyperparameter optimization algorithm that uses evolution strategies to estimate the entire hypergradient (both the direct and indirect component). We use our own implementation for this method.
- AID-CG (Grazzi et al., 2020; Rajeswaran et al., 2019): approximate implicit differentiation with conjugate gradient. We use its implementation provided at `https://github.com/prolearner/hypertorch`
- AID-FP (Grazzi et al., 2020): approximate implicit differentiation with fixed-point. We experimented with its implementation at the repository `https://github.com/prolearner/hypertorch`
- ITD-R (REVERSE) (Franceschi et al., 2017): an iterative differentiation method that computes hypergradients using reverse mode automatic differention (RMAD). We use its implementation provided at `https://github.com/prolearner/hypertorch`.

### C.2    Hyperparameters details for shallow HR experiments in Section 4.1

For the linear embedding case, we set the smoothing parameter $\mu$ to be $0.01$ for ESJ and HOZOG. We use the following hyperparameters for all compared methods. The number of inner GD steps is fixed to $N = 20$ with the learning rate of $\alpha = 0.001$. For the outer optimizer, we use Adam (Kingma & Ba, 2014) with a learning rate of $0.05$. The value of $\gamma$ in eq. (12) is set to be $0.1$. For the two-layer net case, we use $\mu = 0.1$ for ESJ and HOZOG. For all methods, we set $N = 10$, $\alpha = 0.001$, $\beta = 0.001$, and use Adam with a learning rate of $0.01$ as the outer optimizer.

### C.3    Specifications on Baseline Stochastic Algorithms in Section 4.2

We compare our stochastic algorithm ESJ-S with the following baseline stochastic bilevel algorithms.

- stocBiO: an approximate implicit differentiation method that uses Neumann Series to estimate the Hessian inverse. We use its implementation available at `https://github.com/JunjieYang97/StocBio`.
- AID-CG-S and AID-FP-S: stochastic versions of AID-CG and AID-FP, respectively. We use their implementations in the repository `https://github.com/prolearner/hypertorch`.

### C.4    Specifications for Few-shot Meta-Learning in Section 4.3

The miniImageNet dataset (Vinyals et al., 2016) is a large-scale benchmark for few-shot learning generated from ImageNet (Russakovsky et al., 2015) Russakovsky. The dataset consists of 100 classes with each class containing 600 images of size 84 × 84. Following (Arnold et al., 2019), we split the classes into 64 classes for meta-training, 16 classes for meta-validation, and 20 classes for meta-testing. More specfically, we use 20000 tasks for meta-training, 600 tasks for meta-validation, and 600 tasks for meta-testing. We normalize all image pixels by their means and standard deviations over RGB channels and do not perform any additional data augmentation. At each meta-iteration, we sample a batch of 16 training tasks and update the parameters based on these tasks. We set the smoothness parameter to be $\mu = 0.1$ and use $N = 30$ inner steps. We use SGD with a learning rate of $\alpha = 0.01$ as inner optimizer and Adam with a learning rate of $\beta = 0.01$ as outer (meta) optimizer.

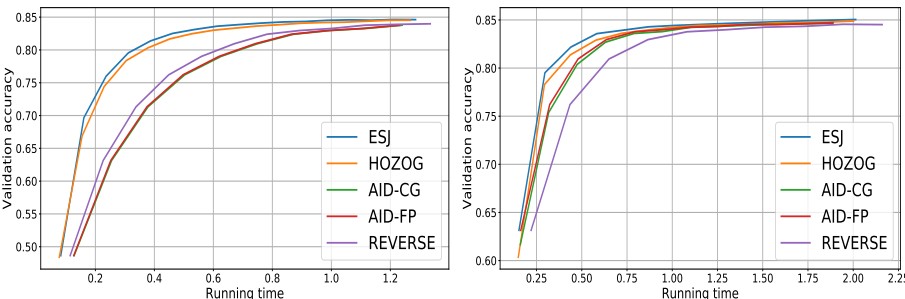

Figure 4: Classification results on 20 Newsgroup dataset. **Left:** number of inner GD step $N = 5$. **Right:** number of inner GD steps $N = 10$. Running time is in seconds.

## D  EXPERIMENTS ON HYPERPARAMETER OPTIMIZATION

Hyperparameter optimization (HO) is the problem of finding the set of the best hyperparamters (either representational or regularization parameters) that yield the optimal value of some criterion of model quality (usually a validation loss on unseen data). HO can be posed as a bilevel optimization problem in which the inner problem corresponds to finding the model parameters by minimizing a training loss (usually regularized) for the given hyperparameters and then the outer problem minimizes over the hyperparameters. Hence, HO can be mathematically expressed as follows

$$
\begin{aligned}
\min_{\lambda} \ & \mathcal{L}_{\mathrm{val}}(\lambda) := \tfrac{1}{|\mathcal{D}_{\mathrm{val}}|} \sum_{\xi \in \mathcal{D}_{\mathrm{val}}} \mathcal{L}\left(w^*(\lambda); \xi\right) \\
\text{s.t.} \ \ & w^*(\lambda) = \arg\min_{w} \mathcal{L}_{\mathrm{tr}}(w, \lambda) := \tfrac{1}{|\mathcal{D}_{\mathrm{tr}}|} \sum_{\zeta \in \mathcal{D}_{\mathrm{tr}}} (\mathcal{L}(w, \lambda; \zeta) + \mathcal{R}(w, \lambda)),
\end{aligned}
\tag{15}
$$

where $\mathcal{L}$ is a loss function (e.g., logistic loss), $\mathcal{R}(w, \lambda)$ is a regularizer, and $\mathcal{D}_{tr}$ and $\mathcal{D}_{val}$ are respectively training and validation data. Note that the loss function used to identify hyperparameters must be different from the one used to find model parameters; otherwise models with higher complexities would be always favored. This is usually achieved in HO by using different data splits (here $\mathcal{D}_{val}$ and $\mathcal{D}_{tr}$) to compute validation and training losses, and by adding a regularizer term on the training loss.

Following (Franceschi et al., 2017; Grazzi et al., 2020), we perform classification on the 20 Newsgroup dataset, where the classifier is modeled by an affine transformation, the cost function $\mathcal{L}$ is the cross-entrpy loss, and $\mathcal{R}(w, \lambda)$ is a strongly-convex regularizer. We set one $l_2$-regularization hyperparameter for each weight in $w$, so that $\lambda$ and $w$ have the same size.

For ESJ and HOZOG, we use GD with a learning rate of $100$ and a momentum of $0.9$ to perform the inner updates. The outer learning rate is set to be $0.02$. We set the smoothing parameter ($\mu$ in Algorithm 1) to be $0.01$. For AID-FP, AID-CG, and REVERSE we use the suggested hyperparameters in their implementations accompanying the paper Grazzi et al. (2020).

It can be seen from Figure 4, our method ESJ slightly outperforms HOZOG and converges faster than the other AID and ITD based approaches. We note that the similar performance for ESJ and HOZOG can be explained by the fact that in HO, the hypergradient expession in eq. (6) contains only the second term (the first term is zero), which is very close to the approximation in HOZOG method. However, as we have seen in the other experiments (not for HO), ESJ is a more robust and stable bilevel optimizer than HOZOG, and it achieves good performance across many bilevel problems.

## E  SUPPORTING TECHNICAL LEMMAS

In this section, we provide auxiliary lemmas that are used for proving the convergence results for the algorithms ESJ and ESJ-S.

In the following proofs, we let $L = \max\{L_g, L_f\}$ and $D$ be such that $\|y^*(x)\| \le D$.

First we recall that for any two matrices $A \in \mathbb{R}^{m \times r}$ and $B \in \mathbb{R}^{r \times n}$, we have the following upper-bound on the Frobenius norm of their product,

$$\|AB\|_F \le \|A\| \|B\|_F. \tag{16}$$

The following lemma follows directly from the Lipschitz properties in Assumptions 2 and 3.

**Lemma 1.** *Suppose that Assumptions 2 and 3 hold. Then, the stochastic derivatives $\nabla F(x,y;\xi)$, $\nabla_x \nabla_y G(x,y;\xi)$, and $\nabla_y^2 G(x,y;\xi)$ have bounded variances, i.e., for any $(x,y)$ and $\xi$ we have:*

- $\mathbb{E}_\xi \|\nabla F(x,y;\xi) - \nabla f(x,y)\|^2 \le M^2$;
- $\mathbb{E}_\xi \|\nabla_x \nabla_y G(x,y;\xi) - \nabla_x \nabla_y g(x,y)\|_F^2 \le L^2$;
- $\mathbb{E}_\xi \|\nabla_y^2 G(x,y;\xi) - \nabla_y^2 g(x,y)\|_F^2 \le L^2$.

Using Lemma 2.2 in Ghadimi & Wang (2018), the following lemma characterizes the Lipschitz property of the gradient of the total objective $\Phi(x) = f(x, y^*(x))$.

**Lemma 2.** *Suppose that Assumptions 1, 2, and 3 hold. Then, we have:*

$$\|\nabla \Phi(x_2) - \nabla \Phi(x_1)\| \le L_\Phi \|x_2 - x_1\| \qquad \forall x_1 \in \mathbb{R}^p, x_2 \in \mathbb{R}^p,$$

*where the constant $L_\Phi = L + \frac{2L^2 + \tau M^2}{\mu_g} + \frac{\rho L M + L^3 + \tau M L}{\mu_g^2} + \frac{\rho L^2 M}{\mu_g^3}$.*

We next provide some essential properties of the ES-based (i.e., zeroth-order) gradient oracle in eq. (4), due to Nesterov & Spokoiny (2017).

**Lemma 3.** *Let $h : \mathbb{R}^n \to \mathbb{R}$ be a differentiable function with L-Lipsctitz gradient. Define its Gaussian smooth approximation $h_\mu(x) = \mathbb{E}_u [h(x + \mu u)]$, where $\mu > 0$ and $u \in \mathbb{R}^n$ is a standard Gaussian random vector. Then, $h_\mu$ is differentiable and we have:*

- *The gradient of $h_\mu$ takes the form*

$$\nabla h_\mu(x) = \mathbb{E}_u \frac{h(x + \mu u) - h(x)}{\mu} u.$$

- *For any $x \in \mathbb{R}^n$,*

$$\|\nabla h_\mu(x) - \nabla h(x)\| \le \frac{\mu^2}{2} L(n+3)^{3/2}.$$

- *For any $x \in \mathbb{R}^n$,*

$$\mathbb{E}_u \left\| \frac{h(x + \mu u) - h(x)}{\mu} u \right\|^2 \le 4(n+4) \|\nabla h_\mu(x)\|^2 + \frac{3}{2} \mu^2 L^2 (n+5)^3.$$

Note the first item in Lemma 3 implies that the oracle in eq. (4) is in indeed an unbiased estimator of the gradient of the smoothed function $h_\mu$.

**Lemma 4.** *Suppose that Assumptions 1 and 2 hold. The Jacobian $\mathcal{J}_* = \frac{\partial y^*(x)}{\partial x}$ has bounded norm:*

$$\|\mathcal{J}_*\|_F \le \frac{L}{\mu_g}. \tag{17}$$

*Proof of Lemma 4.* From the first order optimality condition of $y^*(x)$, we have $\nabla_y g(x, y^*(x)) = 0$. Hence, the Implicit Function Theorem implies:

$$\mathcal{J}_* = - \left[ \nabla_y^2 g(x, y^*(x)) \right]^{-1} \nabla_x \nabla_y g(x, y^*(x)). \tag{18}$$

Taking norms and applying eq. (16) together with Assumptions 1 and 2 yield the desired result

$$\|\mathcal{J}_*\|_F \le \|\nabla_x \nabla_y g(x, y^*(x))\|_F \left\| \left[ \nabla_y^2 g(x, y^*(x)) \right]^{-1} \right\| \le \frac{L}{\mu_g}. \tag{19}$$

$\square$

**Lemma 5.** *Suppose that Assumptions 1 and 2 hold. The Jacobian $\mathcal{J}_N = \frac{\partial y_k^N}{\partial x_k}$ has bounded norm:*

$$\left\| \mathcal{J}_N \right\|_F \leq \frac{L}{\mu_g}. \tag{20}$$

*Proof of Lemma 5.* The inner loop gradient descent updates writes

$$y_k^t = y_k^{t-1} - \alpha \nabla_y g\left(x_k, y_k^{t-1}\right), \quad t = 1, \ldots, N.$$

Taking derivatives w.r.t. $x_k$ yields

$$\mathcal{J}_t = \mathcal{J}_{t-1} - \alpha \nabla_x \nabla_y g\left(x_k, y_k^{t-1}\right) - \alpha \mathcal{J}_{t-1} \nabla_y^2 g\left(x_k, y_k^{t-1}\right)$$
$$= \mathcal{J}_{t-1}\left(I - \alpha \nabla_y^2 g\left(x_k, y_k^{t-1}\right)\right) - \alpha \nabla_x \nabla_y g\left(x_k, y_k^{t-1}\right).$$

Telescoping over $t$ from 1 to $N$ yields

$$\mathcal{J}_N = \mathcal{J}_0 \prod_{t=0}^{N-1}\left(I - \alpha \nabla_y^2 g\left(x_k, y_k^t\right)\right) - \alpha \sum_{t=0}^{N-1} \nabla_x \nabla_y g\left(x_k, y_k^t\right) \prod_{m=t+1}^{N-1}\left(I - \alpha \nabla_y^2 g\left(x_k, y_k^m\right)\right)$$
$$= -\alpha \sum_{t=0}^{N-1} \nabla_x \nabla_y g\left(x_k, y_k^t\right) \prod_{m=t+1}^{N-1}\left(I - \alpha \nabla_y^2 g\left(x_k, y_k^m\right)\right). \tag{21}$$

Hence, we have

$$\left\| \mathcal{J}_N \right\|_F \leq \alpha \sum_{t=0}^{N-1}\left\| \nabla_x \nabla_y g\left(x_k, y_k^t\right) \right\|_F \left\| \prod_{m=t+1}^{N-1}\left(I - \alpha \nabla_y^2 g\left(x_k, y_k^m\right)\right) \right\|$$
$$\stackrel{(i)}{\leq} \alpha \sum_{t=0}^{N-1} L \prod_{m=t+1}^{N-1}\left\| I - \alpha \nabla_y^2 g\left(x_k, y_k^m\right) \right\|$$
$$\stackrel{(ii)}{\leq} \alpha L \sum_{t=0}^{N-1}(1 - \alpha \mu_g)^{N-1-t}$$
$$= \alpha L \sum_{t=0}^{N-1}(1 - \alpha \mu_g)^t \leq \frac{L}{\mu_g}$$

where $(i)$ follows from Assumption 2 and $(ii)$ applies the strong-convexity of function $g(x, \cdot)$. This completes the proof. □

**Lemma 6.** *Suppose that Assumptions 1 and 2 hold. Then, the Jacobian in the stochastic algorithm ESJ-S $\mathcal{J}_N = \frac{\partial Y_k^N}{\partial x_k}$ has bounded norm, as shown below.*

$$\left\| \mathcal{J}_N \right\|_F \leq \frac{L}{\mu_g}. \tag{22}$$

*Proof of Lemma 6.* The proof follows similarly to Lemma 5. □

### E.1 PROOF OF PROPOSITION 1

**Proposition 1** (Re-stated)**.** *Suppose that Assumptions 1 and 2 hold. Define the constant*

$$L_{\mathcal{J}} = \left(1 + \frac{L}{\mu_g}\right)\left(\frac{\tau}{\mu_g} + \frac{\rho L}{\mu_g^2}\right). \tag{23}$$

*Then, the Jacobian $\mathcal{J}_N(x) = \frac{\partial y^N(x)}{\partial x}$ is $L_{\mathcal{J}}$-Lipschitz with respect to $x$ under the Frobenious norm:*

$$\left\| \mathcal{J}_N(x_1) - \mathcal{J}_N(x_2) \right\|_F \leq L_{\mathcal{J}}\left\| x_1 - x_2 \right\| \qquad \forall x_1 \in \mathbb{R}^p, x_2 \in \mathbb{R}^p. \tag{24}$$

*Proof of Proposition 1.* Using eq. (21), we have

$$\mathcal{J}_N(x) = -\alpha \sum_{t=0}^{N-1} \nabla_x \nabla_y g\left(x, y^t(x)\right) \prod_{m=t+1}^{N-1} \left(I - \alpha \nabla_y^2 g\left(x, y^m(x)\right)\right), \quad x \in \mathbb{R}^p$$

Hence, for $x_1 \in \mathbb{R}^p$ and $x_2 \in \mathbb{R}^p$, we have

$$\left\|\mathcal{J}_N(x_1) - \mathcal{J}_N(x_2)\right\|_F$$
$$= \alpha \Big\| \sum_{t=0}^{N-1} \nabla_x \nabla_y g\left(x_1, y^t(x_1)\right) \prod_{m=t+1}^{N-1} \left(I - \alpha \nabla_y^2 g\left(x_1, y^m(x_1)\right)\right)$$
$$- \sum_{t=0}^{N-1} \nabla_x \nabla_y g\left(x_2, y^t(x_2)\right) \prod_{m=t+1}^{N-1} \left(I - \alpha \nabla_y^2 g\left(x_2, y^m(x_2)\right)\right) \Big\|_F$$
$$\leq \alpha \sum_{t=0}^{N-1} \Big\| \nabla_x \nabla_y g\left(x_1, y^t(x_1)\right) \prod_{m=t+1}^{N-1} \left(I - \alpha \nabla_y^2 g\left(x_1, y^m(x_1)\right)\right)$$
$$- \nabla_x \nabla_y g\left(x_2, y^t(x_2)\right) \prod_{m=t+1}^{N-1} \left(I - \alpha \nabla_y^2 g\left(x_2, y^m(x_2)\right)\right) \Big\|_F$$
$$\overset{(i)}{\leq} \alpha \sum_{t=0}^{N-1} \left\| \nabla_x \nabla_y g\left(x_1, y^t(x_1)\right) \right\|_F \left\| A_t(x_1) - A_t(x_2) \right\|$$
$$+ \alpha \sum_{t=0}^{N-1} \left\| A_t(x_2) \right\| \left\| \nabla_x \nabla_y g\left(x_1, y^t(x_1)\right) - \nabla_x \nabla_y g\left(x_2, y^t(x_2)\right) \right\|_F \qquad (25)$$

where we define $A_t(x) = \prod_{m=t+1}^{N-1} \left(I - \alpha \nabla_y^2 g\left(x, y^m(x)\right)\right)$ and $(i)$ follows from eq. (16).

Next we upper bound the quantity $\left\| A_t(x_1) - A_t(x_2) \right\|$, as shown below.

$$\left\| A_t(x_1) - A_t(x_2) \right\| \leq \Big\| \alpha \left( \nabla_y^2 g\left(x_2, y^{t+1}(x_2)\right) - \nabla_y^2 g\left(x_2, y^{t+1}(x_2)\right) \right) A_{t+1}(x_1)$$
$$+ \left(I - \alpha \nabla_y^2 g\left(x_2, y^{t+1}(x_2)\right)\right) \left(A_{t+1}(x_1) - A_{t+1}(x_2)\right) \Big\|$$
$$\leq \alpha \left\| A_{t+1}(x_1) \right\| \left\| \nabla_y^2 g\left(x_1, y^{t+1}(x_1)\right) - \nabla_y^2 g\left(x_2, y^{t+1}(x_2)\right) \right\|$$
$$+ \left\| I - \alpha \nabla_y^2 g\left(x_2, y^{t+1}(x_2)\right) \right\| \left\| A_{t+1}(x_1) - A_{t+1}(x_2) \right\|$$
$$\leq (1 - \alpha \mu_g) \left\| A_{t+1}(x_1) - A_{t+1}(x_2) \right\|$$
$$+ \alpha \rho (1 + \frac{L}{\mu_g})(1 - \alpha \mu_g)^{N-t-2} \left\| x_1 - x_2 \right\|, \qquad (26)$$

where the last inequality follows from Lemma 5 and Assumptions 1 and 2.

Telescoping eq. (26) over $t$ yields

$$\left\| A_t(x_1) - A_t(x_2) \right\| \leq (1 - \alpha \mu_g)^{N-t-2} \left\| A_{N-2}(x_1) - A_{N-2}(x_2) \right\|$$
$$+ \sum_{m=0}^{N-t-3} \alpha \rho (1 + \frac{L}{\mu_g})(1 - \alpha \mu_g)^{N-t-2-m}(1 - \alpha \mu_g)^m \left\| x_1 - x_2 \right\|$$
$$= (1 - \alpha \mu_g)^{N-t-2} \left\| \nabla_y^2 g\left(x_1, y^{N-1}(x_1)\right) - \nabla_y^2 g\left(x_2, y^{N-1}(x_2)\right) \right\|$$
$$+ \sum_{m=0}^{N-t-3} \alpha \rho (1 + \frac{L}{\mu_g})(1 - \alpha \mu_g)^{N-t-2} \left\| x_1 - x_2 \right\|$$
$$\overset{(i)}{\leq} \alpha \rho (1 + \frac{L}{\mu_g})(1 - \alpha \mu_g)^{N-t-2} \left\| x_1 - x_2 \right\|$$
$$+ (N - t - 2) \alpha \rho (1 + \frac{L}{\mu_g})(1 - \alpha \mu_g)^{N-t-2} \left\| x_1 - x_2 \right\|$$

$$=\alpha\rho(1+\frac{L}{\mu_g})(N-t-1)(1-\alpha\mu_g)^{N-t-2}\|x_1-x_2\|, \tag{27}$$

where $(i)$ follows from Lemma 5 and Assumption 2. Replacing eq. (27) in eq. (25) and using Assumption 2, we have

$$\|\mathcal{J}_N(x_1)-\mathcal{J}_N(x_2)\|_F$$

$$\leq \alpha\sum_{t=0}^{N-1} L\alpha\rho(1+\frac{L}{\mu_g})(N-t-1)(1-\alpha\mu_g)^{N-t-2}\|x_1-x_2\|$$

$$+\alpha\sum_{t=0}^{N-1}\tau(1+\frac{L}{\mu_g})(1-\alpha\mu_g)^{N-t-1}\|x_1-x_2\|$$

$$\leq \alpha^2 L\rho(1+\frac{L}{\mu_g})\|x_1-x_2\|\sum_{m=0}^{N-1}m(1-\alpha\mu_g)^{m-1}+\frac{\tau}{\mu_g}(1+\frac{L}{\mu_g})\|x_1-x_2\|$$

$$\leq \frac{\rho L}{\mu_g^2}(1+\frac{L}{\mu_g})\|x_1-x_2\|+\frac{\tau}{\mu_g}(1+\frac{L}{\mu_g})\|x_1-x_2\| \tag{28}$$

where we use $\sum_{m=0}^{N-1}mx^{m-1}\leq\frac{1}{\alpha^2\mu_g^2}$ in eq. (28), which can be obtained by taking derivatives for the expression $\sum_{m=0}^{N-1}x^m$ with respect to $x$. Hence, rearranging and using the definition of $L_{\mathcal{J}}$ in eq. (23) finishes the proof. □

**Lemma 7.** *Suppose that Assumptions 1 and 2 hold. Define the constant*

$$L_{\mathcal{J}} = \left(1+\frac{L}{\mu_g}\right)\left(\frac{\tau}{\mu_g}+\frac{\rho L}{\mu_g^2}\right).$$

*Then, the Jacobian $\mathcal{J}_N(x) = \frac{\partial Y^N(x;\cdot)}{\partial x}$ is $L_{\mathcal{J}}$-Lipschitz with respect to $x$ under the Frobenius norm:*

$$\big\|\mathcal{J}_N(x_1;\cdot)-\mathcal{J}_N(x_2;\cdot)\big\|_F \leq L_{\mathcal{J}}\big\|x_1-x_2\big\| \qquad \forall x_1\in\mathbb{R}^p, x_2\in\mathbb{R}^p.$$

*Proof of Lemma 7.* The proof follows similarly to that for Proposition 1. □

## F  PROOFS FOR DETERMINISTIC BILEVEL OPTIMIZATION

For notation convenience, we define the following quantities:

$$\hat{\mathcal{J}}_{N,j} = \hat{\mathcal{J}}_N(x_k,u_j) = \begin{pmatrix} \frac{y_1^N(x_k+\mu u_j)-y_1^N(x_k)}{\mu}u_j^\top \\ \vdots \\ \frac{y_d^N(x_k+\mu u_j)-y_d^N(x_k)}{\mu}u_j^\top \end{pmatrix}, \quad \mathcal{J}_N = \frac{\partial y_k^N}{\partial x_k}, \quad \mathcal{J}_* = \frac{\partial y_k^*}{\partial x_k}, \tag{29}$$

where $u_j\in\mathbb{R}^p, j=1,\ldots,Q$ are standard Gaussian vectors. Let $y_{i,\mu}^N(x_k)$ be the Gaussian smooth approximation of $y_i^N(x_k)$. We collect $y_{i,\mu}^N(x_k)$ for $i=1,\ldots,d$ together as a vector $y_\mu^N(x_k)$, which is the Gaussian approximation of the vector $y^N(x_k)$. If $\mu>0$, $y_\mu^N(x_k)$ is differentiable and we let $\mathcal{J}_\mu$ be the Jocobian given by

$$\mathcal{J}_\mu = \frac{\partial y_\mu^N(x_k)}{\partial x_k}. \tag{30}$$

We approximate $\frac{\partial y_k^N}{\partial x_k}$ using the average ES estimator given by $\hat{\mathcal{J}}_N = \frac{1}{Q}\sum_{j=1}^{Q}\hat{\mathcal{J}}_{N,j}$. The hypergradient is then approximated as

$$\widehat{\nabla}\Phi(x_k) = \nabla_x f(x_k,y_k^N)+\hat{\mathcal{J}}_N^\top\nabla_y f(x_k,y_k^N)$$

$$= \nabla_x f(x_k,y_k^N)+\frac{1}{Q}\sum_{j=1}^{Q}\hat{\mathcal{J}}_{N,j}^\top\nabla_y f(x_k,y_k^N). \tag{31}$$

Let $\delta_j = \frac{y^N(x_k + \mu u_j) - y^N(x_k)}{\mu}$, and let $\delta_{i,j}$ be the $i$-th component of $\delta_j$. Hence, we have

$$\hat{\mathcal{J}}_{N,j} = \begin{pmatrix} \delta_{1,j} u_j^\top \\ \delta_{2,j} u_j^\top \\ \vdots \\ \delta_{d,j} u_j^\top \end{pmatrix}.$$

$$\hat{\mathcal{J}}_{N,j}^\top \nabla_y f(x_k, y_k^N) = \begin{pmatrix} \delta_{1,j} u_j & \delta_{2,j} u_j & \dots & \delta_{d,j} u_j \end{pmatrix} \nabla_y f(x_k, y_k^N)$$
$$= \langle \delta_j, \nabla_y f(x_k, y_k^N) \rangle u_j. \tag{32}$$

Using eq. (31) and eq. (32), the estimator for the hypergradient can thus be computed as

$$\widehat{\nabla}\Phi(x_k) = \nabla_x f(x_k, y_k^N) + \frac{1}{Q} \sum_{j=1}^{Q} \langle \delta_j, \nabla_y f(x_k, y_k^N) \rangle u_j.$$

### F.1   Proof of Proposition 2

**Proposition 2** (Re-stated). *Suppose that Assumptions 1, 2, and 3 hold. Then, the expected estimation error can be upper-bounded as follows:*

$$\mathbb{E}\big\|\widehat{\nabla}\Phi(x_k) - \nabla\Phi(x_k)\big\|^2 \leq 2L^2 D^2 (1 - \alpha\mu_g)^N + 4\frac{L^4}{\mu_g^2} D^2 (1 - \alpha\mu_g)^N + 24(4p + 15)\frac{L^2 M^2}{Q\mu_g^2}$$
$$+ \frac{\mu^2}{Q} L_{\mathcal{J}}^2 M^2 d\mathcal{P}_4(p) + \frac{24L^2 M^2 (1 - \alpha\mu_g)^{2N}}{\mu_g^2} + 6\mu^2 L_{\mathcal{J}}^2 M^2 d(p + 3)^3$$
$$+ \frac{48M^2 (\tau\mu_g + L\rho)^2}{\mu_g^4} (1 - \alpha\mu_g)^{N-1} D^2, \tag{33}$$

*where the expectation $\mathbb{E}[\cdot]$ is conditioned on $x_k$ and $y_k^N$.*

*Proof of Proposition 2.* Based on the definitions of $\nabla\Phi(x_k)$ and $\widehat{\nabla}\Phi(x_k)$ and conditioning on $x_k$ and $y_k^N$, we have

$$\mathbb{E}\big\|\widehat{\nabla}\Phi(x_k) - \nabla\Phi(x_k)\big\|^2$$
$$\leq 2\big\|\nabla_x f(x_k, y_k^N) - \nabla_x f(x_k, y_k^*)\big\|^2 + 2\mathbb{E}\big\|\hat{\mathcal{J}}_N^\top \nabla_y f(x_k, y_k^N) - \mathcal{J}_*^\top \nabla_y f(x_k, y_k^*)\big\|^2$$
$$\leq 2L^2 \big\|y_k^N - y_k^*\big\|^2 + 4\big\|\mathcal{J}_*\big\|_F^2 \big\|\nabla_y f(x_k, y_k^N) - \nabla_y f(x_k, y_k^*)\big\|^2$$
$$\quad + 4\mathbb{E}\big\|\hat{\mathcal{J}}_N - \mathcal{J}_*\big\|_F^2 \big\|\nabla_y f(x_k, y_k^N)\big\|^2$$
$$\overset{(i)}{\leq} 2L^2 D^2 (1 - \alpha\mu_g)^N + 4\frac{L^4}{\mu_g^2} \big\|y_k^N - y_k^*\big\|^2 + 4M^2 \mathbb{E}\big\|\hat{\mathcal{J}}_N - \mathcal{J}_*\big\|_F^2$$
$$\overset{(ii)}{\leq} 2L^2 D^2 (1 - \alpha\mu_g)^N + 4\frac{L^4}{\mu_g^2} D^2 (1 - \alpha\mu_g)^N + 4M^2 \mathbb{E}\big\|\hat{\mathcal{J}}_N - \mathcal{J}_*\big\|_F^2 \tag{34}$$

where $(i)$ follows from Lemma 4 and Assumption 3, and $(i)$ and $(ii)$ also use the following result for full GD (when applied to a strongly-convex function).

$$\big\|y_k^N - y_k^*\big\|^2 \leq (1 - \alpha\mu_g)^N D^2.$$

Next, we upper-bound the last term $\mathbb{E}\big\|\hat{\mathcal{J}}_N - \mathcal{J}_*\big\|_F^2$ at the last line of eq. (34). First note that

$$\mathbb{E}\big\|\hat{\mathcal{J}}_N - \mathcal{J}_*\big\|_F^2 \leq 3\mathbb{E}\big\|\hat{\mathcal{J}}_N - \mathcal{J}_\mu\big\|_F^2 + 3\big\|\mathcal{J}_N - \mathcal{J}_*\big\|_F^2 + 3\big\|\mathcal{J}_\mu - \mathcal{J}_N\big\|_F^2. \tag{35}$$

We then upper-bound each term of the right hand side of eq. (35). For the first term, we have

$$
\begin{aligned}
\mathbb{E}\big\|\hat{\mathcal{J}}_N - \mathcal{J}_\mu\big\|_F^2 &= \mathbb{E}\big\|\frac{1}{Q}\sum_{j=1}^{Q}\hat{\mathcal{J}}_{N,j} - \mathcal{J}_\mu\big\|_F^2 \\
&= \frac{1}{Q^2}\mathbb{E}\big\|\sum_{j=1}^{Q}\left(\hat{\mathcal{J}}_{N,j} - \mathcal{J}_\mu\right)\big\|_F^2 \\
&= \frac{1}{Q^2}\mathbb{E}\left(\sum_{j=1}^{Q}\big\|\hat{\mathcal{J}}_{N,j} - \mathcal{J}_\mu\big\|_F^2 + 2\sum_{i<j}\left\langle \hat{\mathcal{J}}_{N,i} - \mathcal{J}_\mu, \hat{\mathcal{J}}_{N,j} - \mathcal{J}_\mu \right\rangle\right) \\
&= \frac{1}{Q^2}\sum_{j=1}^{Q}\mathbb{E}\big\|\hat{\mathcal{J}}_{N,j} - \mathcal{J}_\mu\big\|_F^2 \\
&= \frac{1}{Q}\mathbb{E}\big\|\hat{\mathcal{J}}_{N,j} - \mathcal{J}_\mu\big\|_F^2, \quad j \in \{1,\dots,Q\}.
\end{aligned}
\tag{36}
$$

We next upper-bound the term $\mathbb{E}\big\|\hat{\mathcal{J}}_{N,j} - \mathcal{J}_\mu\big\|_F^2$ in eq. (36).

$$
\begin{aligned}
\mathbb{E}\big\|\hat{\mathcal{J}}_{N,j} - \mathcal{J}_\mu\big\|_F^2 &= \mathbb{E}\big\|\hat{\mathcal{J}}_{N,j}\big\|_F^2 - \big\|\mathcal{J}_\mu\big\|_F^2 \tag{37} \\
&\overset{(i)}{\le} \sum_{i=1}^{d}\left(4(p+4)\big\|\nabla y_{i,\mu}^N\big\|^2 + \frac{3}{2}\mu^2 L_\mathcal{J}^2(p+5)^3\right) - \sum_{i=1}^{d}\big\|\nabla y_{i,\mu}^N\big\|^2 \\
&\le \sum_{i=1}^{d}\left((4p+15)\big\|\nabla y_{i,\mu}^N\big\|^2 + \frac{3}{2}\mu^2 L_\mathcal{J}^2(p+5)^3\right), \tag{38}
\end{aligned}
$$

where $(i)$ follows by applying Lemma 3 to the components of vector $y^N(x_k)$ which have Lipschitz gradients by Proposition 1. Then, noting that $\big\|\nabla y_{i,\mu}^N\big\|^2 \le 2\big\|\nabla y_i^N\big\|^2 + \frac{1}{2}\mu^2 L_\mathcal{J}^2(p+3)^3$ and replacing in eq. (38), we have

$$
\begin{aligned}
\mathbb{E}\big\|\hat{\mathcal{J}}_{N,j} - \mathcal{J}_\mu\big\|_F^2 &\le \sum_{i=1}^{d}\left(2(4p+15)\big\|\nabla y_i^N\big\|^2 + \mu^2 L_\mathcal{J}^2 \mathcal{P}_4(p)\right) \\
&\le 2(4p+15)\big\|\mathcal{J}_N\big\|_F^2 + \mu^2 L_\mathcal{J}^2 d\mathcal{P}_4(p) \\
&\overset{(i)}{\le} 2(4p+15)\frac{L^2}{\mu_g^2} + \mu^2 L_\mathcal{J}^2 d\mathcal{P}_4(p), \tag{39}
\end{aligned}
$$

where $(i)$ follows from Lemma 5 and $\mathcal{P}_4$ is a polynomial of degree 4 in $p$. Combining eq. (36) and eq. (38) yields

$$
\mathbb{E}\big\|\hat{\mathcal{J}}_N - \mathcal{J}_\mu\big\|_F^2 \le 2(4p+15)\frac{L^2}{Q\mu_g^2} + \frac{\mu^2}{Q}L_\mathcal{J}^2 d\mathcal{P}_4(p). \tag{40}
$$

We next upper-bound the second term at the right hand side of eq. (35), which can be upper-bounded using eq. (41) in Ji et al. (2021), as shown below.

$$
\big\|\mathcal{J}_N - \mathcal{J}_*\big\|^2 \le \frac{2L^2(1-\alpha\mu_g)^{2N}}{\mu_g^2} + \frac{4(\tau\mu_g + L\rho)^2}{\mu_g^4}(1-\alpha\mu_g)^{N-1}D^2. \tag{41}
$$

We finally upper-bound the last term at the right hand side of eq. (35) using Lemma 3.

$$
\begin{aligned}
\big\|\mathcal{J}_\mu - \mathcal{J}_N\big\|_F^2 &= \sum_{i=1}^{d}\big\|\nabla y_{i,\mu}^N - \nabla y_i^N\big\|^2 \\
&\le \frac{\mu^2}{2}L_\mathcal{J}^2 d(p+3)^3. \tag{42}
\end{aligned}
$$

Substituting eq. (40), eq. (41) and eq. (42) into eq. (35) yields

$$
\begin{aligned}
\mathbb{E}\big\|\hat{\mathcal{J}}_N - \mathcal{J}_*\big\|_F^2 \leq &6(4p+15)\frac{L^2}{Q\mu_g^2} + \frac{\mu^2}{Q}L_{\mathcal{J}}^2 d\mathcal{P}_4(p) + \frac{6L^2(1-\alpha\mu_g)^{2N}}{\mu_g^2} \\
&+ \frac{12(\tau\mu_g + L\rho)^2}{\mu_g^4}(1-\alpha\mu_g)^{N-1}D^2 + \frac{3\mu^2}{2}L_{\mathcal{J}}^2 d(p+3)^3.
\end{aligned}
\tag{43}
$$

Finally, the bound for the expected estimation error in eq. (34) becomes

$$
\begin{aligned}
\mathbb{E}\big\|\widehat{\nabla}\Phi(x_k) - \nabla\Phi(x_k)\big\|^2 \leq &2L^2D^2(1-\alpha\mu_g)^N + 4\frac{L^4}{\mu_g^2}D^2(1-\alpha\mu_g)^N + 24(4p+15)\frac{L^2M^2}{Q\mu_g^2} \\
&+ \frac{\mu^2}{Q}L_{\mathcal{J}}^2 M^2 d\mathcal{P}_4(p) + \frac{24L^2M^2(1-\alpha\mu_g)^{2N}}{\mu_g^2} + 6\mu^2 L_{\mathcal{J}}^2 M^2 d(p+3)^3 \\
&+ \frac{48M^2(\tau\mu_g + L\rho)^2}{\mu_g^4}(1-\alpha\mu_g)^{N-1}D^2.
\end{aligned}
\tag{44}
$$

This completes the proof. □

## F.2 PROOF OF THEOREM 1

**Theorem 1** (Re-stated). *Suppose that Assumptions 1, 2, and 3 hold. Choose the inner- and outer-loop stepsizes respectively as $\alpha \leq \frac{1}{L}$ and $\beta = \frac{1}{4L_\Phi}$, where $L_\Phi = L + \frac{2L^2+\tau M^2}{\mu_g} + \frac{\rho LM + L^3 + \tau ML}{\mu_g^2} + \frac{\rho L^2 M}{\mu_g^3}$. Then, the iterates $x_k$ for $k = 0, ..., K-1$ of ESJ in Algorithm 1 satisfy:*

$$
\frac{1}{K}\sum_{k=0}^{K-1}\mathbb{E}\big\|\nabla\Phi(x_k)\big\|^2 \leq \frac{16L_\phi(\Phi(x_0) - \Phi^*)}{K} + 3\mathcal{D}
\tag{45}
$$

*with $\Phi^* = \inf_x \Phi(x)$ and $\mathcal{D}$ is the upper-bound established in Proposition 2 and is given by*

$$
\begin{aligned}
\mathcal{D} = &2L^2D^2(1-\alpha\mu_g)^N + 4\frac{L^4}{\mu_g^2}D^2(1-\alpha\mu_g)^N + 24(4p+15)\frac{L^2M^2}{Q\mu_g^2} \\
&+ \frac{\mu^2}{Q}L_{\mathcal{J}}^2 M^2 d\mathcal{P}_4(p) + \frac{24L^2M^2(1-\alpha\mu_g)^{2N}}{\mu_g^2} + 6\mu^2 L_{\mathcal{J}}^2 M^2 d(p+3)^3 \\
&+ \frac{48M^2(\tau\mu_g + L\rho)^2}{\mu_g^4}(1-\alpha\mu_g)^{N-1}D^2.
\end{aligned}
\tag{46}
$$

*Proof of Theorem 1.* Using the Lipschitzness of function $\Phi(x_k)$, we have

$$
\begin{aligned}
\Phi(x_{k+1}) \leq &\Phi(x_k) + \langle\nabla\Phi(x_k), x_{k+1} - x_k\rangle + \frac{L_\phi}{2}\big\|x_{k+1} - x_k\big\|^2 \\
\leq &\Phi(x_k) - \beta\langle\nabla\Phi(x_k), \widehat{\nabla}\Phi(x_k)\rangle + \frac{L_\phi}{2}\beta^2\big\|\widehat{\nabla}\Phi(x_k)\big\|^2 \\
\leq &\Phi(x_k) - \beta\langle\nabla\Phi(x_k), \widehat{\nabla}\Phi(x_k) - \nabla\Phi(x_k)\rangle - \beta\big\|\nabla\Phi(x_k)\big\|^2 \\
&+ L_\phi\beta^2\left(\big\|\nabla\Phi(x_k)\big\|^2 + \big\|\widehat{\nabla}\Phi(x_k) - \nabla\Phi(x_k)\big\|^2\right) \\
\leq &\Phi(x_k) - (\frac{\beta}{2} - L_\phi\beta^2)\big\|\nabla\Phi(x_k)\big\|^2 + (\frac{\beta}{2} + L_\phi\beta^2)\big\|\widehat{\nabla}\Phi(x_k) - \nabla\Phi(x_k)\big\|^2.
\end{aligned}
\tag{47}
$$

Let $\mathbb{E}_k[\cdot] = \mathbb{E}_{u_{k,1:Q}}[\cdot|x_k, y_k^N]$ be the expectation over the Gaussian vectors $u_{k,1}, \ldots, u_{k,Q}$ conditioned on $x_k$ and $y_k^N$. Applying the expectation $\mathbb{E}_k[\cdot]$ to eq. (47) yields

$$
\begin{aligned}
\mathbb{E}_k\Phi(x_{k+1}) \leq &\Phi(x_k) - (\frac{\beta}{2} - L_\phi\beta^2)\big\|\nabla\Phi(x_k)\big\|^2 + (\frac{\beta}{2} + L_\phi\beta^2)\mathbb{E}_k\big\|\widehat{\nabla}\Phi(x_k) - \nabla\Phi(x_k)\big\|^2 \\
\leq &\Phi(x_k) - (\frac{\beta}{2} - L_\phi\beta^2)\big\|\nabla\Phi(x_k)\big\|^2 + (\frac{\beta}{2} + L_\phi\beta^2)\mathcal{D},
\end{aligned}
\tag{48}
$$

where $\mathcal{D}$ represent the upper-bound established for the estimation error in Proposition 2. Now taking total expectation over $\mathcal{U}_k = \{u_{1,1:Q}, \ldots, u_{k,1:Q}\}$, we have

$$E_{k+1} \leq E_k - (\frac{\beta}{2} - L_\phi \beta^2)\mathbb{E}_{\mathcal{U}_k}\left\|\nabla\Phi(x_k)\right\|^2 + (\frac{\beta}{2} + L_\phi\beta^2)\mathcal{D} \tag{49}$$

where $E_k = \mathbb{E}_{\mathcal{U}_{k-1}}\Phi(x_k)$. Summing up the inequalities in eq. (49) for $k = 0, \ldots, K-1$ yields

$$E_k \leq E_0 - (\frac{\beta}{2} - L_\phi \beta^2)\sum_{k=0}^{K-1}\mathbb{E}_{\mathcal{U}_k}\left\|\nabla\Phi(x_k)\right\|^2 + (\frac{\beta}{2} + L_\phi\beta^2)K\mathcal{D}. \tag{50}$$

Setting $\beta = \frac{1}{4L_\phi}$, denoting by $\Phi^* = \inf_x \Phi(x)$, and rearranging eq. (50), we have

$$\frac{1}{K}\sum_{k=0}^{K-1}\mathbb{E}_{\mathcal{U}_k}\left\|\nabla\Phi(x_k)\right\|^2 \leq \frac{16L_\phi(\Phi(x_0) - \Phi^*)}{K} + 3\mathcal{D}. \tag{51}$$

Hence, the proof is finished. $\qquad\square$

# G    PROOFS FOR STOCHASTIC BILEVEL OPTIMIZATION

Define the following quantities

$$\hat{\mathcal{J}}_{N,j} = \hat{\mathcal{J}}_N(x_k, u_j) = \begin{pmatrix} \frac{Y_1^N(x_k+\mu u_j;\mathcal{S})-Y_1^N(x_k;\mathcal{S})}{\mu}u_j^\top \\ \vdots \\ \frac{Y_d^N(x_k+\mu u_j;\mathcal{S})-Y_d^N(x_k;\mathcal{S})}{\mu}u_j^\top \end{pmatrix}, \quad \mathcal{J}_N = \frac{\partial Y_k^N}{\partial x_k}, \quad \mathcal{J}_* = \frac{\partial y_k^*}{\partial x_k}$$

where $u_j \in \mathbb{R}^p, j = 1, \ldots, Q$ are standard Gaussian vectors and $Y_k^N$ is the output of SGD obtained with the minibatches $\{\mathcal{S}_0, ..., \mathcal{S}_{N-1}\}$.

Conditioning on $x_k$ and $Y_k^N$ and taking expectation over $u_j$ yields

$$\mathbb{E}_{u_j}\hat{\mathcal{J}}_{N,j} = \mathbb{E}_{u_j}\begin{pmatrix} \frac{Y_1^N(x_k+\mu u_j;\mathcal{S})-Y_1^N(x_k;\mathcal{S})}{\mu}u_j^\top \\ \vdots \\ \frac{Y_d^N(x_k+\mu u_j;\mathcal{S})-Y_d^N(x_k;\mathcal{S})}{\mu}u_j^\top \end{pmatrix}$$
$$= \begin{pmatrix} \nabla_x^\top Y_{1,\mu}^N(x_k;\mathcal{S}) \\ \vdots \\ \nabla_x^\top Y_{d,\mu}^N(x_k;\mathcal{S}) \end{pmatrix}$$
$$= \mathcal{J}_\mu(\mathcal{S})$$

where $Y_{i,\mu}^N(x_k;\mathcal{S})$ is the $i$-th component of vector $Y_\mu^N(x_k;\mathcal{S})$, which is the entry-wise Gaussian smooth approximation of vector $Y^N(x_k;\mathcal{S})$. Let $\mathbb{E}_k[\cdot] = \mathbb{E}[\cdot|x_k, Y_k^N] = \mathbb{E}_{\mathcal{D}_F, u_{1:q}}$ be the expectation over the Gaussian vectors and the sample minibatch $\mathcal{D}_F$ conditioned on $x_k$ and $Y_k^N$.

## G.1    PROOF OF PROPOSITION 3

**Proposition 3** (Re-stated). *Suppose that Assumptions 1, 2, and 4 hold. Choose the inner-loop stepsize as $\alpha = \frac{2}{L+\mu_g}$. Define the constants*

$$C_\gamma = (1-\alpha\mu_g)\left(1 - \alpha\mu_g + \frac{\alpha}{\gamma} + \frac{\alpha L}{\gamma\mu_g}\right), C_{xy} = \alpha\left(\alpha + \gamma(1-\alpha\mu_g) + \alpha\frac{L}{\mu_g}\right), C_y = \frac{L}{\mu_g}C_{xy}$$

$$\Gamma = 2(\tau^2 C_{xy} + \rho^2 C_y)\frac{\sigma^2}{\mu_g LS} + 2\frac{L^2}{S}(C_{xy} + C_y), \quad \lambda = 2(\tau^2 C_{xy} + \rho^2 C_y)D^2, \tag{52}$$

where $\gamma$ is such that $\gamma \geq \frac{L+\mu_g}{\mu_g^2}$. Then, we have:

$$\mathbb{E}\big\|\mathcal{J}_N - \mathcal{J}_*\big\|_F^2 \leq C_\gamma^N \frac{L^2}{\mu_g^2} + \frac{\lambda(L+\mu_g)^2(1-\alpha\mu_g)C_\gamma^{N-1}}{(L+\mu_g)^2(1-\alpha\mu_g) - (L-\mu_g)^2} + \frac{\Gamma}{1-C_\gamma}.$$

*Proof of Proposition 3.* Based on the SGD updates, we have

$$Y_k^t = Y_k^{t-1} - \alpha\nabla_y G\left(x_k, Y_k^{t-1}; \mathcal{S}_{t-1}\right), \quad t = 1, \ldots, N$$

Taking the derivatives w.r.t. $x_k$ yields

$$\begin{aligned}
\mathcal{J}_t =& \mathcal{J}_{t-1} - \alpha\nabla_x\nabla_y G\left(x_k, Y_k^{t-1}; \mathcal{S}_{t-1}\right) - \alpha\mathcal{J}_{t-1}\nabla_y^2 G\left(x_k, Y_k^{t-1}; \mathcal{S}_{t-1}\right) \\
\mathcal{J}_t - \mathcal{J}_* =& \mathcal{J}_{t-1} - \mathcal{J}_* - \alpha\nabla_x\nabla_y G\left(x_k, Y_k^{t-1}; \mathcal{S}_{t-1}\right) - \alpha\mathcal{J}_{t-1}\nabla_y^2 G\left(x_k, Y_k^{t-1}; \mathcal{S}_{t-1}\right) \\
& + \alpha\left(\nabla_x\nabla_y g\left(x_k, y_k^*\right) + \mathcal{J}_*\nabla_y^2 g\left(x_k, y_k^*\right)\right) \\
=& \mathcal{J}_{t-1} - \mathcal{J}_* - \alpha\left(\nabla_x\nabla_y G\left(x_k, Y_k^{t-1}; \mathcal{S}_{t-1}\right) - \nabla_x\nabla_y g\left(x_k, y_k^*\right)\right) \\
& - \alpha\left(\mathcal{J}_{t-1} - \mathcal{J}_*\right)\nabla_y^2 G\left(x_k, Y_k^{t-1}; \mathcal{S}_{t-1}\right) \\
& + \alpha\mathcal{J}_*\left(\nabla_y^2 g\left(x_k, y_k^*\right) - \nabla_y^2 G\left(x_k, Y_k^{t-1}; \mathcal{S}_{t-1}\right)\right).
\end{aligned}$$

Hence, using the triangle inequality, we have

$$\begin{aligned}
\big\|\mathcal{J}_t - \mathcal{J}_*\big\|_F \overset{(i)}{\leq}& \big\|\left(\mathcal{J}_{t-1} - \mathcal{J}_*\right)\left(I - \nabla_y^2 G\left(x_k, Y_k^{t-1}; \mathcal{S}_{t-1}\right)\right)\big\|_F \\
& + \alpha\big\|\nabla_x\nabla_y G\left(x_k, Y_k^{t-1}; \mathcal{S}_{t-1}\right) - \nabla_x\nabla_y g\left(x_k, y_k^*\right)\big\|_F \\
& + \alpha\big\|\mathcal{J}_*\left(\nabla_y^2 G\left(x_k, Y_k^{t-1}; \mathcal{S}_{t-1}\right) - \nabla_y^2 g\left(x_k, y_k^*\right)\right)\big\|_F,
\end{aligned}$$

where $(i)$ follows from Assumption 1. We then further have

$$\begin{aligned}
\big\|\mathcal{J}_t - \mathcal{J}_*\big\|_F^2 \leq& \\
& (1-\alpha\mu_g)^2\big\|\mathcal{J}_{t-1} - \mathcal{J}_*\big\|_F^2 + \alpha^2\big\|\nabla_x\nabla_y G\left(x_k, Y_k^{t-1}; \mathcal{S}_{t-1}\right) - \nabla_x\nabla_y g\left(x_k, y_k^*\right)\big\|_F^2 \\
& + \alpha^2\frac{L^2}{\mu_g^2}\big\|\nabla_y^2 G\left(x_k, Y_k^{t-1}; \mathcal{S}_{t-1}\right) - \nabla_y^2 g\left(x_k, y_k^*\right)\big\|_F^2 \\
& + 2\alpha(1-\alpha\mu_g)\underbrace{\big\|\mathcal{J}_{t-1} - \mathcal{J}_*\big\|_F\big\|\nabla_x\nabla_y G\left(x_k, Y_k^{t-1}; \mathcal{S}_{t-1}\right) - \nabla_x\nabla_y g\left(x_k, y_k^*\right)\big\|_F}_{P_1} \\
& + 2\alpha(1-\alpha\mu_g)\frac{L}{\mu_g}\underbrace{\big\|\mathcal{J}_{t-1} - \mathcal{J}_*\big\|_F\big\|\nabla_y^2 G\left(x_k, Y_k^{t-1}; \mathcal{S}_{t-1}\right) - \nabla_y^2 g\left(x_k, y_k^*\right)\big\|_F}_{P_2} \\
& + 2\alpha^2\frac{L}{\mu_g}\underbrace{\big\|\nabla_y^2 G\left(x_k, Y_k^{t-1}; \mathcal{S}_{t-1}\right) - \nabla_y^2 g\left(x_k, y_k^*\right)\big\|_F\big\|\nabla_x\nabla_y G\left(x_k, Y_k^{t-1}; \mathcal{S}_{t-1}\right) - \nabla_x\nabla_y g\left(x_k, y_k^*\right)\big\|_F}_{P_3}.
\end{aligned}$$

The terms $P_1$, $P_2$ and $P_3$ in the above inequality can be transformed as follows using the Peter-Paul version of Young's inequality.

$$\begin{aligned}
P_1 \leq& \frac{1}{2\gamma}\big\|\mathcal{J}_{t-1} - \mathcal{J}_*\big\|_F^2 + \frac{\gamma}{2}\big\|\nabla_x\nabla_y G\left(x_k, Y_k^{t-1}; \mathcal{S}_{t-1}\right) - \nabla_x\nabla_y g\left(x_k, y_k^*\right)\big\|_F^2, \quad \gamma > 0 \\
P_2 \leq& \frac{1}{2\gamma}\big\|\mathcal{J}_{t-1} - \mathcal{J}_*\big\|_F^2 + \frac{\gamma}{2}\big\|\nabla_y^2 G\left(x_k, Y_k^{t-1}; \mathcal{S}_{t-1}\right) - \nabla_y^2 g\left(x_k, y_k^*\right)\big\|_F^2, \quad \gamma > 0 \\
P_3 \leq& \frac{1}{2}\big\|\nabla_y^2 G\left(x_k, Y_k^{t-1}; \mathcal{S}_{t-1}\right) - \nabla_y^2 g\left(x_k, y_k^*\right)\big\|_F^2 \\
& + \frac{1}{2}\big\|\nabla_x\nabla_y G\left(x_k, Y_k^{t-1}; \mathcal{S}_{t-1}\right) - \nabla_x\nabla_y g\left(x_k, y_k^*\right)\big\|_F^2
\end{aligned}$$

Note that the trade-off constant $\gamma$ controls the contraction coefficient (factor in front of $\big\|\mathcal{J}_{t-1} - \mathcal{J}_*\big\|_F^2$). Hence, we have

$$\big\|\mathcal{J}_t - \mathcal{J}_*\big\|_F^2$$

$$\leq \left( (1-\alpha\mu_g)^2 + \frac{\alpha}{\gamma}(1-\alpha\mu_g) + \frac{\alpha L}{\gamma\mu_g}(1-\alpha\mu_g) \right) \left\| \mathcal{J}_{t-1} - \mathcal{J}_* \right\|_F^2$$
$$+ \left( \alpha^2 + \alpha\gamma(1-\alpha\mu_g) + \alpha^2\frac{L}{\mu_g} \right) \left\| \nabla_x\nabla_y G\left(x_k, Y_k^{t-1}; \mathcal{S}_{t-1}\right) - \nabla_x\nabla_y g\left(x_k, y_k^*\right) \right\|_F^2$$
$$+ \left( \alpha^2\frac{L^2}{\mu_g^2} + \alpha\gamma\frac{L}{\mu_g}(1-\alpha\mu_g) + \alpha^2\frac{L}{\mu_g} \right) \left\| \nabla_y^2 G\left(x_k, Y_k^{t-1}; \mathcal{S}_{t-1}\right) - \nabla_y^2 g\left(x_k, y_k^*\right) \right\|_F^2.$$

Let $\mathbb{E}_{t-1}[\cdot] = \mathbb{E}[\cdot | x_k, Y_k^{t-1}]$. Conditioning on $x_k$ and $Y_k^{t-1}$ and taking expectations yield

$$\mathbb{E}_{t-1}\left\| \mathcal{J}_t - \mathcal{J}_* \right\|_F^2 \leq$$
$$C_\gamma\left\| \mathcal{J}_{t-1} - \mathcal{J}_* \right\|_F^2 + C_{xy}\mathbb{E}_{t-1}\left\| \nabla_x\nabla_y G\left(x_k, Y_k^{t-1}; \mathcal{S}_{t-1}\right) - \nabla_x\nabla_y g\left(x_k, y_k^*\right) \right\|_F^2$$
$$+ C_y\mathbb{E}_{t-1}\left\| \nabla_y^2 G\left(x_k, Y_k^{t-1}; \mathcal{S}_{t-1}\right) - \nabla_y^2 g\left(x_k, y_k^*\right) \right\|_F^2, \tag{53}$$

where $C_\gamma$, $C_{xy}$ and $C_y$ are defined as follows

$$C_\gamma = (1-\alpha\mu_g)\left(1-\alpha\mu_g + \frac{\alpha}{\gamma} + \frac{\alpha L}{\gamma\mu_g}\right), C_{xy} = \alpha\left(\alpha + \gamma(1-\alpha\mu_g) + \alpha\frac{L}{\mu_g}\right), C_y = \frac{L}{\mu_g}C_{xy}.$$

Conditioning on $x_k$ and $Y_k^{t-1}$, we have

$$\mathbb{E}_{t-1}\left\| \nabla_x\nabla_y G\left(x_k, Y_k^{t-1}; \mathcal{S}_{t-1}\right) - \nabla_x\nabla_y g\left(x_k, y_k^*\right) \right\|_F^2$$
$$\leq 2\mathbb{E}_{t-1}\left\| \nabla_x\nabla_y g\left(x_k, Y_k^{t-1}\right) - \nabla_x\nabla_y g\left(x_k, y_k^*\right) \right\|_F^2$$
$$+ 2\mathbb{E}_{t-1}\left\| \nabla_x\nabla_y G\left(x_k, Y_k^{t-1}; \mathcal{S}_{t-1}\right) - \nabla_x\nabla_y g\left(x_k, Y_k^{t-1}\right) \right\|_F^2$$
$$\overset{(i)}{\leq} 2\frac{L^2}{S} + 2\tau^2\left\| Y_k^{t-1} - y_k^* \right\|^2, \tag{54}$$

where $(i)$ follows from Lemma 1 and Assumption 2. Similarly we can derive

$$\mathbb{E}_{t-1}\left\| \nabla_y^2 G\left(x_k, Y_k^{t-1}; \mathcal{S}_{t-1}\right) - \nabla_y^2 g\left(x_k, y_k^*\right) \right\|_F^2 \leq 2\frac{L^2}{S} + 2\rho^2\left\| Y_k^{t-1} - y_k^* \right\|^2. \tag{55}$$

Combining eq. (53), eq. (54), and eq. (55) we obtain

$$\mathbb{E}_{t-1}\left\| \mathcal{J}_t - \mathcal{J}_* \right\|_F^2 \leq C_\gamma\left\| \mathcal{J}_{t-1} - \mathcal{J}_* \right\|_F^2 + 2(\tau^2 C_{xy} + \rho^2 C_y)\left\| Y_k^{t-1} - y_k^* \right\|^2$$
$$+ 2\frac{L^2}{S}(C_{xy} + C_y). \tag{56}$$

Unconditioning on $x_k$ and $Y_k^{t-1}$ and taking total expectations of eq. (56) yield

$$\mathbb{E}\left\| \mathcal{J}_t - \mathcal{J}_* \right\|_F^2 \leq C_\gamma\mathbb{E}\left\| \mathcal{J}_{t-1} - \mathcal{J}_* \right\|_F^2 + 2(\tau^2 C_{xy} + \rho^2 C_y)\mathbb{E}\left\| Y_k^{t-1} - y_k^* \right\|^2 + 2\frac{L^2}{S}(C_{xy} + C_y)$$
$$\overset{(i)}{\leq} C_\gamma\mathbb{E}\left\| \mathcal{J}_{t-1} - \mathcal{J}_* \right\|_F^2 + 2(\tau^2 C_{xy} + \rho^2 C_y)\left( \left(\frac{L-\mu_g}{L+\mu_g}\right)^{2(t-1)}D^2 + \frac{\sigma^2}{\mu_g LS} \right)$$
$$+ 2\frac{L^2}{S}(C_{xy} + C_y),$$

where $(i)$ follows from the analysis of SGD for a strongly-convex function. Let $\Gamma = 2(\tau^2 C_{xy} + \rho^2 C_y)\frac{\sigma^2}{\mu_g LS} + 2\frac{L^2}{S}(C_{xy} + C_y)$ and $\lambda = 2(\tau^2 C_{xy} + \rho^2 C_y)D^2$. Then, we have

$$\mathbb{E}\left\| \mathcal{J}_t - \mathcal{J}_* \right\|_F^2 \leq C_\gamma\mathbb{E}\left\| \mathcal{J}_{t-1} - \mathcal{J}_* \right\|_F^2 + \lambda\left(\frac{L-\mu_g}{L+\mu_g}\right)^{2(t-1)} + \Gamma. \tag{57}$$

Telescoping eq. (57) over $t$ from $N$ down to 1 yields

$$\mathbb{E}\left\| \mathcal{J}_N - \mathcal{J}_* \right\|_F^2 \leq C_\gamma^N\mathbb{E}\left\| \mathcal{J}_0 - \mathcal{J}_* \right\|_F^2 + \lambda\sum_{t=0}^{N-1}\left(\frac{L-\mu_g}{L+\mu_g}\right)^{2t}C_\gamma^{N-1-t} + \Gamma\sum_{t=0}^{N-1}C_\gamma^t \tag{58}$$

which, in conjunction with $\left(\frac{L-\mu_g}{L+\mu_g}\right)^2 \leq 1 - \alpha\mu_g$ and $\gamma \geq \frac{L+\mu_g}{\mu_g^2}$ such that $C_\gamma \leq 1 - \alpha\mu_g$, yields

$$
\mathbb{E}\big\|\mathcal{J}_N - \mathcal{J}_*\big\|_F^2 \leq C_\gamma^N \frac{L^2}{\mu_g^2} + \lambda C_\gamma^{N-1} \sum_{t=0}^{N-1} \left(\frac{(L-\mu_g)^2}{(L+\mu_g)^2(1-\alpha\mu_g)}\right)^t + \frac{\Gamma}{1-C_\gamma}
$$

$$
\leq C_\gamma^N \frac{L^2}{\mu_g^2} + \frac{\lambda(L+\mu_g)^2(1-\alpha\mu_g)C_\gamma^{N-1}}{(L+\mu_g)^2(1-\alpha\mu_g) - (L-\mu_g)^2} + \frac{\Gamma}{1-C_\gamma}. \tag{59}
$$

The proof is then complete. $\qquad\qquad\qquad\qquad\qquad\qquad\qquad\qquad\qquad\qquad\qquad\qquad\qquad\square$

**Lemma 8.** *Suppose that Assumptions 1, 2, 3, and 4 hold. Set the inner-loop stepsize as $\alpha = \frac{2}{L+\mu_g}$. Then, we have*

$$
\mathbb{E}\big\|\mathbb{E}_k\widehat{\nabla}\Phi(x_k) - \nabla\Phi(x_k)\big\|^2
$$

$$
\leq 8M^2\left(C_\gamma^N \frac{L^2}{\mu_g^2} + \frac{\lambda(L+\mu_g)^2(1-\alpha\mu_g)C_\gamma^{N-1}}{(L+\mu_g)^2(1-\alpha\mu_g) - (L-\mu_g)^2} + \frac{\Gamma}{1-C_\gamma} + \frac{\mu^2}{2}L_{\mathcal{J}}^2 d(p+3)^3\right)
$$

$$
+ 2L^2\left(1 + 2\frac{L^2}{\mu_g^2}\right)\left(\left(\frac{L-\mu_g}{L+\mu_g}\right)^{2N} D^2 + \frac{\sigma^2}{\mu_g LS}\right), \tag{60}
$$

*where the expectation $\mathbb{E}_k[\cdot]$ is conditioned on $x_k$ and $Y_k^N$.*

*Proof of Lemma 8.* Conditioning on $x_k$ and $Y_k^N$, we have

$$
\mathbb{E}_k\widehat{\nabla}\Phi(x_k) = \nabla_x f(x_k, Y_k^N) + \mathcal{J}_\mu^\top \nabla_y f(x_k, Y_k^N).
$$

Recall $\nabla\Phi(x_k) = \nabla_x f(x_k, y_k^*) + \mathcal{J}_*^\top \nabla_y f(x_k, y_k^*)$. Thus, we have

$$
\big\|\mathbb{E}_k\widehat{\nabla}\Phi(x_k) - \nabla\Phi(x_k)\big\|^2
$$

$$
\leq 2\big\|\nabla_x f(x_k, Y_k^N) - \nabla_x f(x_k, y_k^*)\big\|^2 + 2\big\|\mathcal{J}_\mu^\top \nabla_y f(x_k, Y_k^N) - \mathcal{J}_*^\top \nabla_y f(x_k, y_k^*)\big\|^2
$$

$$
\leq 2L^2\big\|Y_k^N - y_k^*\big\|^2 + 4\big\|\mathcal{J}_\mu^\top \nabla_y f(x_k, Y_k^N) - \mathcal{J}_*^\top \nabla_y f(x_k, Y_k^N)\big\|^2
$$

$$
+ 4\big\|\mathcal{J}_*^\top \nabla_y f(x_k, Y_k^N) - \mathcal{J}_*^\top \nabla_y f(x_k, y_k^*)\big\|^2
$$

$$
\overset{(i)}{\leq} 2L^2\big\|Y_k^N - y_k^*\big\|^2 + 4M^2\big\|\mathcal{J}_\mu - \mathcal{J}_*\big\|_F^2 + 4\frac{L^2}{\mu_g^2}\big\|\nabla_y f(x_k, Y_k^N) - \nabla_y f(x_k, y_k^*)\big\|^2
$$

$$
\leq 2L^2\big\|Y_k^N - y_k^*\big\|^2 + 8M^2\big\|\mathcal{J}_\mu - \mathcal{J}_N\big\|_F^2 + 8M^2\big\|\mathcal{J}_N - \mathcal{J}_*\big\|_F^2 + 4\frac{L^4}{\mu_g^2}\big\|Y_k^N - y_k^*\big\|^2.
$$

where $(i)$ applies Lemma 4 and Assumption 3. Taking expectation of the above inequality yields

$$
\mathbb{E}\big\|\mathbb{E}_k\widehat{\nabla}\Phi(x_k) - \nabla\Phi(x_k)\big\|^2 \leq 2L^2\left(1 + 2\frac{L^2}{\mu_g^2}\right)\mathbb{E}\big\|Y_k^N - y_k^*\big\|^2 + 8M^2\mathbb{E}\big\|\mathcal{J}_N - \mathcal{J}_*\big\|_F^2
$$

$$
+ 8M^2\mathbb{E}\big\|\mathcal{J}_\mu - \mathcal{J}_N\big\|_F^2. \tag{61}
$$

Using the fact that $Y_i^N(x_k; \cdot)$ has Lipschitz gradient (Proposition 1) and Lemma 3, the last term at the right hand side of eq. (61) can be directly upper-bounded as

$$
\big\|\mathcal{J}_\mu - \mathcal{J}_N\big\|_F^2 = \sum_{i=1}^d \big\|\nabla Y_{i,\mu}^N(x_k; \mathcal{S}) - \nabla Y_i^N(x_k; \mathcal{S})\big\|^2 \leq \frac{\mu^2}{2}L_{\mathcal{J}}^2 d(p+3)^3, \tag{62}
$$

where $L_{\mathcal{J}}$ is the Lipschitz constant of the Jacobian $\mathcal{J}_N$ (and also of its rows $\nabla Y_i^N(x_k; \mathcal{S})$) as defined in Proposition 1. Combining eq. (61), eq. (62), Proposition 3, and SGD analysis (as in eq. (60) in Ji et al. (2021)) yields

$$
\mathbb{E}\big\|\mathbb{E}_k\widehat{\nabla}\Phi(x_k) - \nabla\Phi(x_k)\big\|^2
$$

$$\leq 8M^2 \left( C_\gamma^N \frac{L^2}{\mu_g^2} + \frac{\lambda(L+\mu_g)^2(1-\alpha\mu_g)C_\gamma^{N-1}}{(L+\mu_g)^2(1-\alpha\mu_g)-(L-\mu_g)^2} + \frac{\Gamma}{1-C_\gamma} + \frac{\mu^2}{2}L_{\mathcal{J}}^2 d(p+3)^3 \right)$$

$$+ 2L^2 \left(1 + 2\frac{L^2}{\mu_g^2}\right)\left(\left(\frac{L-\mu_g}{L+\mu_g}\right)^{2N}D^2 + \frac{\sigma^2}{\mu_g LS}\right). \tag{63}$$

This finishes the proof. □

### G.2 PROOF OF PROPOSITION 4

**Proposition 4** (Re-stated). *Suppose that Assumptions 1, 2, 3, and 4 hold. Set the inner-loop stepsize as $\alpha = \frac{2}{L+\mu_g}$. Then, we have:*

$$\mathbb{E}\big\|\widehat{\nabla}\Phi(x_k) - \nabla\Phi(x_k)\big\|^2 \leq \Delta + \mathcal{B}_1 \tag{64}$$

*where $\Delta = 8M^2\left(\left(1+\frac{1}{D_f}\right)\frac{4p+15}{Q} + \frac{1}{D_f}\right)\frac{L^2}{\mu_g^2} + 2\frac{M^2}{D_f} + \left(1+\frac{1}{D_f}\right)\frac{4M^2}{Q}\mu^2 dL_{\mathcal{J}}^2\mathcal{P}_4(p) + \frac{4M^2}{D_f}\mu^2 dL_{\mathcal{J}}^2\mathcal{P}_3(p)$*

*and $\mathcal{B}_1$ respresents the upper bound established in Lemma 8.*

*Proof of Proposition 4.* We have, conditioning on $x_k$ and $Y_k^N$

$$\mathbb{E}_k\big\|\widehat{\nabla}\Phi(x_k) - \nabla\Phi(x_k)\big\|^2 = \mathbb{E}_k\big\|\widehat{\nabla}\Phi(x_k) - \mathbb{E}_k\widehat{\nabla}\Phi(x_k)\big\|^2 + \big\|\mathbb{E}_k\widehat{\nabla}\Phi(x_k) - \nabla\Phi(x_k)\big\|^2. \tag{65}$$

Our next step is to upper-bound the first term in eq. (65).

$$\mathbb{E}_k\big\|\widehat{\nabla}\Phi(x_k) - \mathbb{E}_k\widehat{\nabla}\Phi(x_k)\big\|^2$$

$$\leq 2\mathbb{E}_k\big\|\nabla_x F(x_k, Y_k^N; \mathcal{D}_F) - \nabla_x f(x_k, Y_k^N)\big\|^2$$

$$+ 2\mathbb{E}_k\big\|\hat{\mathcal{J}}_N^\top \nabla_y F(x_k, Y_k^N; \mathcal{D}_F) - \mathcal{J}_\mu^\top \nabla_y f(x_k, Y_k^N)\big\|^2$$

$$\leq 2\frac{M^2}{D_f} + 4\mathbb{E}_k\big\|\nabla_y F(x_k, Y_k^N; \mathcal{D}_F)\big\|^2\big\|\hat{\mathcal{J}}_N - \mathcal{J}_\mu\big\|_F^2$$

$$+ 4\mathbb{E}_k\big\|\mathcal{J}_\mu\big\|_F^2\big\|\nabla_y F(x_k, Y_k^N; \mathcal{D}_F) - \nabla_y f(x_k, Y_k^N)\big\|^2$$

$$\leq 2\frac{M^2}{D_f} + 4M^2\left(1 + \frac{1}{D_f}\right)\mathbb{E}_k\big\|\hat{\mathcal{J}}_N - \mathcal{J}_\mu\big\|_F^2 + 4\frac{M^2}{D_f}\big\|\mathcal{J}_\mu\big\|_F^2, \tag{66}$$

where the last two steps follows from Lemma 1.

Next, we upper-bound the term $\mathbb{E}_k\big\|\hat{\mathcal{J}}_N - \mathcal{J}_\mu\big\|_F^2$.

$$\mathbb{E}_k\big\|\hat{\mathcal{J}}_N - \mathcal{J}_\mu\big\|_F^2 = \frac{1}{Q}\mathbb{E}_k\big\|\hat{\mathcal{J}}_{N,j} - \mathcal{J}_\mu\big\|_F^2, \quad j \in \{1, \ldots, Q\}$$

$$\leq \frac{1}{Q}\left(\mathbb{E}_k\big\|\hat{\mathcal{J}}_{N,j}\big\|_F^2 - \big\|\mathcal{J}_\mu\big\|_F^2\right)$$

$$\leq \frac{1}{Q}\sum_{i=1}^d \left(\mathbb{E}_k\Big\|\frac{Y_i^N(x_k + \mu u_j; \mathcal{S}) - Y_i^N(x_k; \mathcal{S})}{\mu}u_j\Big\|^2 - \big\|\nabla Y_{i,\mu}^N(x_k; \mathcal{S})\big\|^2\right). \tag{67}$$

Recall that for a function $h$ with $L$-Lipschitz gradient, we have

$$\mathbb{E}_u\Big\|\frac{h(x+\mu u) - h(x)}{\mu}u\Big\|^2 \leq 4(p+4)\big\|\nabla h_\mu(x)\big\|^2 + \frac{3}{2}\mu^2 L^2(p+5)^3. \tag{68}$$

Then, applying eq. (68) to function $Y_i^N(\cdot; \mathcal{S})$ yields

$$\mathbb{E}_{u_j}\Big\|\frac{Y_i^N(x_k + \mu u_j; \mathcal{S}) - Y_i^N(x_k; \mathcal{S})}{\mu}u_j\Big\|^2$$

$$\leq 4(p+4)\big\|\nabla Y_{i,\mu}^N(x_k; \mathcal{S})\big\|^2 + \frac{3}{2}\mu^2 L_{\mathcal{J}}^2(p+5)^3.$$

Hence, eq. (67) becomes

$$\mathbb{E}_k \big\| \hat{\mathcal{J}}_N - \mathcal{J}_\mu \big\|^2 \leq \frac{4p+15}{Q} \sum_{i=1}^{d} \big\| \nabla Y_{i,\mu}^N(x_k; \mathcal{S}) \big\|^2 + \frac{3\mu^2 dL_{\mathcal{J}}^2}{2Q}(p+5)^3$$

$$\leq \frac{2(4p+15)}{Q} \sum_{i=1}^{d} \big\| \nabla Y_i^N(x_k; \mathcal{S}) \big\|^2 + \frac{\mu^2 dL_{\mathcal{J}}^2}{Q} \mathcal{P}_4(p)$$

$$\leq \frac{2(4p+15)}{Q} \big\| \mathcal{J}_N \big\|_F^2 + \frac{\mu^2 dL_{\mathcal{J}}^2}{Q} \mathcal{P}_4(p), \tag{69}$$

where $\mathcal{P}_4$ is a polynomial of degree $4$ in $p$. Combining eq. (66), eq. (69) and Lemma 3 yields

$$\mathbb{E}_k \big\| \widehat{\nabla}\Phi(x_k) - \mathbb{E}_k \widehat{\nabla}\Phi(x_k) \big\|^2 \leq 2\frac{M^2}{D_f} + 4M^2 \Big(1 + \frac{1}{D_f}\Big)\Big(\frac{2(4p+15)}{Q}\big\|\mathcal{J}_N\big\|_F^2 + \frac{\mu^2 dL_{\mathcal{J}}^2}{Q}\mathcal{P}_4(p)\Big)$$

$$+ \frac{4M^2}{D_f}\Big(2\big\|\mathcal{J}_N\big\|_F^2 + \mu^2 dL_{\mathcal{J}}^2 \mathcal{P}_3(p)\Big)$$

$$\leq 8M^2\Big(\Big(1 + \frac{1}{D_f}\Big)\frac{4p+15}{Q} + \frac{1}{D_f}\Big)\frac{L^2}{\mu_g^2} + 2\frac{M^2}{D_f}$$

$$+ \Big(1 + \frac{1}{D_f}\Big)\frac{4M^2}{Q}\mu^2 dL_{\mathcal{J}}^2 \mathcal{P}_4(p) + \frac{4M^2}{D_f}\mu^2 dL_{\mathcal{J}}^2 \mathcal{P}_3(p)$$

$$= \Delta, \tag{70}$$

where $\Delta = 8M^2\Big(\Big(1 + \frac{1}{D_f}\Big)\frac{4p+15}{Q} + \frac{1}{D_f}\Big)\frac{L^2}{\mu_g^2} + 2\frac{M^2}{D_f} + \Big(1 + \frac{1}{D_f}\Big)\frac{4M^2}{Q}\mu^2 dL_{\mathcal{J}}^2 \mathcal{P}_4(p) + \frac{4M^2}{D_f}\mu^2 dL_{\mathcal{J}}^2 \mathcal{P}_3(p)$.

Taking total expectations of both eq. (65) and eq. (70) and combining, we have

$$\mathbb{E}\big\| \widehat{\nabla}\Phi(x_k) - \nabla\Phi(x_k) \big\|^2 = \mathbb{E}\big\| \widehat{\nabla}\Phi(x_k) - \mathbb{E}_k\widehat{\nabla}\Phi(x_k) \big\|^2 + \mathbb{E}\big\| \mathbb{E}_k\widehat{\nabla}\Phi(x_k) - \nabla\Phi(x_k) \big\|^2$$

$$\leq \Delta + \mathcal{B}_1 \tag{71}$$

where $\mathcal{B}_1$ representing the upper-bound established in eq. (63). The proof is then complete. $\quad\square$

### G.3    PROOF OF THEOREM 2

**Theorem 2** (Re-stated)**.** *Suppose that Assumptions 1, 2, 3, and 4 hold. Set the inner- and outer-loop stepsizes respectivelly as $\alpha = \frac{2}{L+\mu_g}$ and $\beta = \frac{1}{4L_\Phi}$, where $L = \max\{L_f, L_g\}$ and the constant $L_\Phi$ is defined as in Theorem 1. Then, the iterates $x_k, k = 0, ..., K-1$ of the ESJ-S algorithm satisfy*

$$\frac{1}{K}\sum_{k=0}^{K-1} \mathbb{E}\big\| \nabla\Phi(x_k) \big\|^2 \leq \frac{16(\Phi(x_0) - \Phi^*)L_\Phi}{K} + 3\mathcal{B}_1 + \Delta, \tag{72}$$

*where $\Delta$ and $\mathcal{B}_1$ have the following forms*

$$\Delta = 8M^2\Big(\Big(1 + \frac{1}{D_f}\Big)\frac{4p+15}{Q} + \frac{1}{D_f}\Big)\frac{L^2}{\mu_g^2} + 2\frac{M^2}{D_f} + \Big(1 + \frac{1}{D_f}\Big)\frac{4M^2}{Q}\mu^2 dL_{\mathcal{J}}^2 \mathcal{P}_4(p)$$

$$+ \frac{4M^2}{D_f}\mu^2 dL_{\mathcal{J}}^2 \mathcal{P}_3(p)$$

$$\mathcal{B}_1 = 8M^2\Big(C_\gamma^N \frac{L^2}{\mu_g^2} + \frac{\lambda(L+\mu_g)^2(1-\alpha\mu_g)C_\gamma^{N-1}}{(L+\mu_g)^2(1-\alpha\mu_g) - (L-\mu_g)^2} + \frac{\Gamma}{1-C_\gamma} + \frac{\mu^2}{2}L_{\mathcal{J}}^2 d(p+3)^3\Big)$$

$$+ 2L^2\Big(1 + 2\frac{L^2}{\mu_g^2}\Big)\Big(\Big(\frac{L-\mu_g}{L+\mu_g}\Big)^{2N} D^2 + \frac{\sigma^2}{\mu_g LS}\Big)$$

*and the constants $\Gamma$, $\lambda$, and $C_\gamma$ are defined in Proposition 3.*

*Proof of Theorem 2.* Using the Lipschitzness of function $\Phi(x_k)$, we have

$$\Phi(x_{k+1}) \leq \Phi(x_k) + \langle \nabla\Phi(x_k), x_{k+1} - x_k \rangle + \frac{L_\phi}{2}\big\| x_{k+1} - x_k \big\|^2$$

$$\leq \Phi(x_k) - \beta \langle \nabla \Phi(x_k), \widehat{\nabla} \Phi(x_k) \rangle + \frac{L_\phi}{2} \beta^2 \big\| \widehat{\nabla} \Phi(x_k) - \nabla \Phi(x_k) + \nabla \Phi(x_k) \big\|^2$$

$$\leq \Phi(x_k) - \beta \langle \nabla \Phi(x_k), \widehat{\nabla} \Phi(x_k) \rangle + L_\phi \beta^2 \left( \big\| \nabla \Phi(x_k) \big\|^2 + \big\| \widehat{\nabla} \Phi(x_k) - \nabla \Phi(x_k) \big\|^2 \right)$$

Hence, taking expectation over the above inequality yields

$$\mathbb{E}\Phi(x_{k+1}) \leq \mathbb{E}\Phi(x_k) - \beta \mathbb{E}\langle \nabla \Phi(x_k), \widehat{\nabla} \Phi(x_k) \rangle + L_\phi \beta^2 \mathbb{E}\big\| \nabla \Phi(x_k) \big\|^2$$
$$+ L_\phi \beta^2 \mathbb{E}\big\| \widehat{\nabla} \Phi(x_k) - \nabla \Phi(x_k) \big\|^2. \tag{73}$$

Also, we have

$$-\mathbb{E}\langle \nabla \Phi(x_k), \widehat{\nabla} \Phi(x_k) \rangle = - \mathbb{E}\langle \nabla \Phi(x_k), \widehat{\nabla} \Phi(x_k) - \nabla \Phi(x_k) \rangle - \mathbb{E}\big\| \nabla \Phi(x_k) \big\|^2$$
$$= \mathbb{E}\left[ -\langle \nabla \Phi(x_k), \mathbb{E}_k \widehat{\nabla} \Phi(x_k) - \nabla \Phi(x_k) \rangle \right] - \mathbb{E}\big\| \nabla \Phi(x_k) \big\|^2$$
$$\leq \mathbb{E}\left( \frac{1}{2} \big\| \nabla \Phi(x_k) \big\|^2 + \frac{1}{2} \big\| \mathbb{E}_k \widehat{\nabla} \Phi(x_k) - \nabla \Phi(x_k) \big\|^2 \right) - \mathbb{E}\big\| \nabla \Phi(x_k) \big\|^2$$
$$= \frac{1}{2} \mathbb{E}\big\| \mathbb{E}_k \widehat{\nabla} \Phi(x_k) - \nabla \Phi(x_k) \big\|^2 - \frac{1}{2} \mathbb{E}\big\| \nabla \Phi(x_k) \big\|^2,$$

which, in conjunction with eq. (73), yields

$$\mathbb{E}\Phi(x_{k+1}) \leq \mathbb{E}\Phi(x_k) + \frac{\beta}{2} \mathbb{E}\big\| \mathbb{E}_k \widehat{\nabla} \Phi(x_k) - \nabla \Phi(x_k) \big\|^2 - \left( \frac{\beta}{2} - L_\phi \beta^2 \right) \mathbb{E}\big\| \nabla \Phi(x_k) \big\|^2$$
$$+ L_\phi \beta^2 \mathbb{E}\big\| \widehat{\nabla} \Phi(x_k) - \nabla \Phi(x_k) \big\|^2. \tag{74}$$

Setting $\beta = \frac{1}{4L_\Phi}$ and using the bounds established in Lemma 8 and Proposition 4, we have

$$\mathbb{E}\Phi(x_{k+1}) \leq \mathbb{E}\Phi(x_k) + \frac{\beta}{2}\mathcal{B}_1 - \frac{\beta}{4}\mathbb{E}\big\| \nabla \Phi(x_k) \big\|^2 + \frac{\beta}{4}(\mathcal{B}_1 + \Delta).$$

Summing up the above inequality over $k$ from $k = 0$ to $k = K - 1$ yields

$$\mathbb{E}\Phi(x_K) \leq \mathbb{E}\Phi(x_0) + \frac{\beta}{2}K\mathcal{B}_1 - \frac{\beta}{4} \sum_{k=0}^{K-1} \mathbb{E}\big\| \nabla \Phi(x_k) \big\|^2 + \frac{\beta}{4}K(\mathcal{B}_1 + \Delta).$$

Rearranging the above inequality yields

$$\frac{1}{K} \sum_{k=0}^{K-1} \mathbb{E}\big\| \nabla \Phi(x_k) \big\|^2 \leq \frac{4\left( \mathbb{E}\Phi(x_0) - \mathbb{E}\Phi(x_K) \right)}{\beta K} + 3\mathcal{B}_1 + \Delta$$
$$\leq \frac{16\left( \Phi(x_0) - \Phi^* \right)L_\Phi}{K} + 3\mathcal{B}_1 + \Delta.$$

The proof is then complete. $\qquad\square$

