# OpenReview forum: "ES-Based Jacobian Enables Faster Bilevel Optimization"
_ICLR.cc/2022/Conference — ICLR 2022 Submitted_

### Official Review · Reviewer_Ws7X · 2021-11-01

**Correctness:** 4
**Technical Novelty And Significance:** 2
**Empirical Novelty And Significance:** 2
**Recommendation:** 3
**Confidence:** 4

**Main Review:**

**Things I like**: The paper proposes an alternative approach to bypass the need of computing the costly second-order information in bilevel optimization and provides theoretical guarantees for the proposed algorithms. The problem is important, and the algorithms are well-motivated.

**Things can be improved**:

**Limited technical novelty**: While this paper uses ES in a more careful manner in the bilevel optimization, it should be noted that the idea of using ES mainly follows from [Gu et al 2021] and the convergence analysis of the bilevel part mainly follows from [Ji et al 2021].

Specifically, compared with [Gu et al 2021], the proposed algorithm only uses ES on Jacobian in (3) not the entire bilevel gradient. On the theoretical side, it is known in optimization literature that applying an ES method will only introduce a small bias that is proportional to \mu, and the remaining of the analysis follows directly from the bilevel optimization [Ji et al 2021]. Also see a couple of earlier and important works such as

[1] Ghadimi, S. and Lan, G., 2013. Stochastic first-and zeroth-order methods for nonconvex stochastic programming. SIAM Journal on Optimization, 23(4), pp.2341-2368.

Building upon previous results in bilevel optimization and ES, the theoretical contribution of this work amounts to establish the estimation error of the proposed gradient estimator, which looks not as impressive as claimed in the introduction.

**High actual complexity**: While the goal is to reduce the complexity of bilevel optimization, the proposed algorithm still incurs large computation overhead. By carefully looking at Theorems 1 and 2, to guarantee convergence, the theory requires the number of queries Q and the batch size S to grow with $\epsilon^-1$, which makes the theory less favorable to practitioners. Such high complexity seems to be hidden in the simulations by choosing small Q and S, which makes the comparison with baselines less convincing.

**Limited empirical verification**: One should provide a thorough empirical studies on the proposed algorithms and the existing ones. However, most of experiments are carried out on simple problems with small networks. And the running time (in second, minute, or hour) is not clearly marked. I have several questions:

i) The baselines on MNIST only get a test accuracy of 93, which is too small. Have the authors tuned the parameters well? To my knowledge, it is not hard to achieve 97 or 98 test accuracies on reasonable networks.

ii) It looks like that the authors do not scale up the learning rate with e.g., 0.1, after several epochs. This learning rate trick is useful in practice, and some algorithm benefits significantly from this trick.

iii) ResNet12 is not big enough to justify the claim on scalability.

As a matter of fact, only a subset of experiments of [Gu et al 2021] has been tested with ESJ using synthetic data in this paper. This seems implausible to claim its practical advantage to HOZOG [Gu et al 2021].


**Summary Of The Paper:**

This paper studies the evolution strategies (ES) based method to estimate the Jacobian matrix used in bilevel optimization. The main idea is to use first-order information to estimate the second-order information in the Jacobian matrix. Some convergence results have been provided, companied with some simulations.

**Summary Of The Review:**

Overall, first-order bilevel optimization is a worthy direction to pursue, but efforts on both technical and empirical novelty are needed to make this paper competitive.

---

> ### Author Response · Authors · 2021-11-23
> **Continued response...**
>
> Q3: The baselines on MNIST only get a test accuracy of 93, which is too small. Have the authors tuned the parameters well? To my knowledge, it is not hard to achieve 97 or 98 test accuracies on reasonable networks.
>
> A3: Thanks for the question! Please note that the baselines here are existing bilevel algorithms, not the standard single-phased algorithm of training a classifier on the MNIST dataset (i.e., the problem setting is different here). The unsatisfactory performance (93% test accuracy) of bilevel baselines is caused by the challenges in handling this bilevel structure, e.g., computation of the hypergradient. We have done a thorough hyparameter selection for these baselines and still they can only reach 93. In fact, as we mentioned in Section 4.2, only our method was able to recover the accuracy of standard single-phased training.
>
> Q4: It looks like that the authors do not scale up the learning rate with e.g., 0.1, after several epochs. This learning rate trick is useful in practice, and some algorithm benefits significantly from this trick.
>
> A4: Thanks! We assume that the reviewer meant scale “down” the learning rate rather than scale “up”. We did try some tricks of decaying learning rate in all our experiments and have used it whenever it can help.
>
> Q5: ResNet12 is not big enough to justify the claim on scalability.
>
> A5: Thanks! In our revision, instead of stating “large neural networks”, we have specified “ResNet-12” to be precise. Please also note that, for few-shot meta-learning, ResNet-12 is already the largest network that has been trained in the literature (see, e.g., [Lee 19’], [Liu 21’]). The networks typically considered in most of few-shot meta-learning literature such as in (MAML [Finn 17’] and ANIL [Raghu 19’]) are much smaller (e.g., 4-layer CNN for miniImageNet dataset and OmniglotNet for Omniglot dataset). And all experiments in all bilevel studies in the literature train much smaller networks as well. The reason for this is due to the nested hierarchy of the problem, which makes it hard to scale to the extremely large networks employed in other machine learning problems.
>
> References:
> [Finn 17’] Chelsea Finn, Pieter Abbeel, and Sergey Levine. Model-agnostic meta-learning for fast adaptation of deep networks. In Proc. International Conference on Machine Learning (ICML), 2017.
>
> [Raghu 19’] Aniruddh Raghu, Maithra Raghu, Samy Bengio, and Oriol Vinyals. Rapid learning or feature reuse? towards understanding the effectiveness of MAML. International Conference on Learning Rep- resentations (ICLR), 2019.
>
> [Lee 19’] Kwonjoon Lee, Subhransu Maji, Avinash Ravichandran, and Stefano Soatto.  Meta-learning with
> differentiable convex optimization.  In Proceedings of the IEEE/CVF Conference on Computer Vision and Pattern Recognition, 2019.
>
> [Liu 21’] Risheng Liu, Yaohua Liu,  Shangzhi Zeng, Jin Zhang. Towards Gradient-based Bilevel Optimization with Non-convex Followers and Beyond. NeurIPS 2021.
>
> Q6: As a matter of fact, only a subset of experiments of [Gu et al 2021] has been tested with ESJ using synthetic data in this paper. This seems implausible to claim its practical advantage to HOZOG [Gu et al 2021].
>
> A6: We have added more experiments on hyperparameter optimization on real datasets in the supplementary materials (see Appendix D), where our proposed method does outperform HOZOG [Gu et al 2021]. We will try to include more experiments in our future revision. Another point we want to mention is that in our meta-learning and hyper-representation experiments (with real datasets) beyond hyperparamter optimization, HOZOG simply fails to convergence (we have tried our best to tune their algorithm but it still fails), and this is why we do not include it in the comparisons in some experiments.

---

> ### Author Response · Authors · 2021-11-23
> **Thank you for your review**
>
> Many thanks for providing the review! In our revision of the paper, we added new experiments in Figures 1 and 3 in Section 4 and in Appendix D, and made various revisions throughout the paper based on all reviewers’ comments. All our changes are highlighted with blue-colored texts. New comments on these changes are very welcome!
>
> Q1: While this paper uses ES in a more careful manner in the bilevel optimization, it should be noted that the idea of using ES mainly follows from [Gu et al 2021] and the convergence analysis of the bilevel part mainly follows from [Ji et al 2021].
>
> Specifically, compared with [Gu et al 2021], the proposed algorithm only uses ES on Jacobian in (3) not the entire bilevel gradient. On the theoretical side, it is known in optimization literature that applying an ES method will only introduce a small bias that is proportional to \mu, and the remaining of the analysis follows directly from the bilevel optimization [Ji et al 2021]. Also see a couple of earlier and important works such as
>
> [1] Ghadimi, S. and Lan, G., 2013. Stochastic first-and zeroth-order methods for nonconvex stochastic programming. SIAM Journal on Optimization, 23(4), pp.2341-2368.
>
> Building upon previous results in bilevel optimization and ES, the theoretical contribution of this work amounts to establish the estimation error of the proposed gradient estimator, which looks not as impressive as claimed in the introduction.
>
> A1: We don’t see the points that the reviewer tried to make here.
>
> Regarding ES method, although the idea was first introduced by [Gu et al 2021] to bilevel optimization (which we stated a few times in the paper), Gu’s algorithm does not perform well, particularly not scalable to even relatively small neural networks. This is why we develop our algorithm using ES method in a different and more thoughtful way. Clearly, our algorithm outperforms Gu’s algorithm significantly. We view  this to be a good contribution that distinguishes our work from Gu’s work.
>
> Regarding theoretical analysis, our approach is not a simple adaptation from [Ji et al 2021]. We hope the reviewer will agree with careful checking through our technical proofs. Here are some key points that are unique to our work:
>
> -For the deterministic case: In Proposition 1, we establish the Lipschitzness of the approximate Jacobian and provide its Lipchitz constant, which is a new result that has not been explored before. Due to the $N$ GD steps in the inner loop, the approximate Jacobian has a nested and recursive structure which makes it mathematically hard to find its Lipchitz constant. This is an essential result as it allows one to use the theoretical results obtained for zeroth-order methods for L-smooth functions [Nesterov 2011’], [Ghadimi 2013’] . This serves as the pillar of showing the convergence guarantee for both the deterministic and stochastic settings.
>
> -For the stochastic case: In Proposition 3, we upper-bound the second moment of the estimation error between the approximate Jacobian and the true Jacobian. Here also several mathematical challenges do arise due to the nested structure of the Jacobian. In fact, to obtain an upper-bound that does not blow up as N increases, one needs to come up with more judicious telescoping coefficient. This essential result also serves as the pillar of showing the convergence guarantee for the stochastic setting.
>
> Q2: While the goal is to reduce the complexity of bilevel optimization, the proposed algorithm still incurs large computation overhead. By carefully looking at Theorems 1 and 2, to guarantee convergence, the theory requires the number of queries Q and the batch size S to grow with , which makes the theory less favorable to practitioners. Such high complexity seems to be hidden in the simulations by choosing small Q and S, which makes the comparison with baselines less convincing.
>
> A2: We want to clarify that our theory is to establish the worst-case performance guarantee for the proposed method. However, the hyperparameter configurations and setups for the worst-case scenarios are unnecessary to give a good performance for many practical experiments. This is why many methods with worst-case performance guarantee will need a careful hyperparameter selection.
>
> Regarding baselines, in all our experiments, we use grid search to find the best parameters for all comparison methods. The comparison is fair.

---

### Official Review · Reviewer_Tjy4 · 2021-11-02

**Correctness:** 3
**Technical Novelty And Significance:** 2
**Empirical Novelty And Significance:** 1
**Recommendation:** 3
**Confidence:** 4

**Main Review:**

Pros:
* The idea of using ES based method to approximate the response Jacobian matrix is interesting.
* Overall, the paper is easy to follow. In particular, the main contribution section makes it easier to highlight the important contribution made in the paper and the related work is written well and makes the algorithm well motivated.

Concerns:
* I believe that the paper misses several important related works. There have been earlier works on using ES-based methods for bilevel optimization (e.g. [1]) that share a similar idea of approximating the Jacobian of the response function. Moreover, STNs ([2], [3]) use the ES gradient approximation method to approximate the response function (or response Jacobian) in an amortized manner. I believe that these should be discussed in the related work section and contribution section of the paper, and should be compared against in the experiment section since the core idea is very similar.
* The key concern about the paper is the lack of rigorous experimentation to show that the proposed method is superior to other bilevel optimization approaches. The paper majorly focuses on the hyper-representation experiments and compares the proposed method to only three baselines (e.g. HOZOG, AID-CG, AID-FP). I believe that the paper should experiment with the algorithm in more diverse settings (e.g. hyperparameter optimization) and compare it to other gradient-based methods (e.g. [2], [3], [5] and various ITD based methods). At the moment, I am not convinced that the proposed algorithm is superior to other bilevel optimizers (both in terms of accuracy and computation it required).
* In the first experiment, the authors do not properly explain why there is an improvement in the outer data accuracy. Does this mean that the proposed method approximates the response Jacobian more accurately than CG?
* What are additional hyperparameters introduced by ESJ-S? I believe that they are Q (number of Gaussian samples) and N. It would be good to have an ablation study of how Q affects the quality of the algorithm.
* In the first experiment, the authors mention that they follow the same experimental procedure as [4], but I couldn’t find the paper. I believe that there are typos in the reference.
* Some mathematical notations are not properly defined throughout the paper and make the paper hard to follow in terms of the notation. For example, in Algorithm 1, the authors did not define variables such as K, N, and Q.

Some minor comments:
* In section “related work”, the paragraph header “Bilevel optimization” is missing “.” at the end.
* In algorithm 1, it would be helpful to identify variables such as K, Q
* In section 3.2, “Differently from …” → “Different from …”
* I believe that some information in the Appendix is outdated. For example, in Appendix B, the authors say “paper demonstrates the superior performance of the proposed ES-based bilevel optimizer in meta-learning and hyperparameter optimization”. However, there was no experiment on hyperparameter optimization.

[1] Sinha, Ankur, Pekka Malo, and Kalyanmoy Deb. "An improved bilevel evolutionary algorithm based on quadratic approximations." 2014 IEEE congress on evolutionary computation (CEC). IEEE, 2014.

[2] MacKay, Matthew, et al. "Self-tuning networks: Bilevel optimization of hyperparameters using structured best-response functions." arXiv preprint arXiv:1903.03088 (2019).

[3] Bae, Juhan, and Roger Grosse. "Delta-STN: Efficient Bilevel Optimization for Neural Networks using Structured Response Jacobians." arXiv preprint arXiv:2010.13514 (2020).

[4] Riccardo Grazzi, Luca Franceschi, Massimiliano Pontil, and Saverio Salzo. Optimizing millions of hyperparameters by implicit differentiation. International Conference on Machine Learning (ICML)), 2020.

[5] Pedregosa, Fabian. "Hyperparameter optimization with approximate gradient." International conference on machine learning. PMLR, 2016.

**Summary Of The Paper:**

The paper aims to reduce the computational cost in solving a bilevel problem. In bilevel optimization, the outer gradient can be decomposed into direct and indirect gradients. The indirect gradient requires computing the response Jacobian, which involves computing & inverting the Hessian and computing mixed derivative. To avoid expensive second-order approximations, the authors propose to use the evolution strategies (ES) based method to approximate the response Jacobian. The authors provide the convergence guarantee of the algorithm and empirically demonstrate that the proposed algorithm performs well on large-scale bilevel problems.

**Summary Of The Review:**

The paper is easy to follow and I believe that the contents of the paper are technically correct. However, the paper doesn't discuss several important related works and the empirical analysis is not rigorous: the authors conduct the empirical investigation in a limited setting and don't compare the proposed method with competitive baselines. It is still unclear if the proposed algorithm is superior to the previous approaches. I believe that there should more in-depth empirical analysis to make the proposed method more convincing.

---

> ### Author Response · Authors · 2021-11-23
> **Continued response...**
>
> Q4: What are additional hyperparameters introduced by ESJ-S? I believe that they are Q (number of Gaussian samples) and N. It would be good to have an ablation study of how Q affects the quality of the algorithm.
>
> A4: Good point! Actually, we did try our method with different values of $Q$ but it turns out that $Q = 1$ always exhibits the best performance. In fact, the batch size $Q$ was introduced only to guarantee theoretical convergence in the worst case, whereas most practical problems do not fall into the worst case, and hence favor much more flexible parameter choices to achieve better performance. This is actually in favor of practitioners as small $Q$ would result in much less computational burden.
>
> Q5: In the first experiment, the authors mention that they follow the same experimental procedure as [4], but I couldn’t find the paper. I believe that there are typos in the reference.
>
> A5: Thanks! The correct reference is:
>
> [Grazzi 20’] Riccardo Grazzi, Luca Franceschi, Massimiliano Pontil, and Saverio Salzo. On the Iteration Complexity of Hypergradient Computation. International Conference on Machine Learning (ICML)), 2020.
>
> Q6: Some mathematical notations are not properly defined throughout the paper and make the paper hard to follow in terms of the notation. For example, in Algorithm 1, the authors did not define variables such as K, N, and Q.
>
> A6: Thanks! We have fixed that.
>
> Some minor comments:
> ·       In section “related work”, the paragraph header “Bilevel optimization” is missing “.” at the end.
> ·       In algorithm 1, it would be helpful to identify variables such as K, Q
> ·       In section 3.2, “Differently from …” → “Different from …”
> ·       I believe that some information in the Appendix is outdated. For example, in Appendix B, the authors say “paper demonstrates the superior performance of the proposed ES-based bilevel optimizer in meta-learning and hyperparameter optimization”. However, there was no experiment on hyperparameter optimization.
>
> Thanks! We have fixed that.

---

> ### Author Response · Authors · 2021-11-23
> **Thank you for your review**
>
> Many thanks for providing the review! In our revision of the paper, we added new experiments in Figures 1 and 3 in Section 4 and in Appendix D, and made various revisions throughout the paper based on all reviewers’ comments. All our changes are highlighted with blue-colored texts. New comments on these changes are very welcome!
>
> Q1. I believe that the paper misses several important related works. There have been earlier works on using ES-based methods for bilevel optimization (e.g. [1]) that share a similar idea of approximating the Jacobian of the response function. Moreover, STNs ([2], [3]) use the ES gradient approximation method to approximate the response function (or response Jacobian) in an amortized manner. I believe that these should be discussed in the related work section and contribution section of the paper, and should be compared against in the experiment section since the core idea is very similar.
>
> A1: Thanks! The purpose of this work is mainly to demonstrate the competitiveness of the proposed ES Jacobian approach as an alternative to existing “generic” methods (such as AID and ITD methods) to estimate the hypergradient. While we find the STNs very interesting, we believe that these are “not generic” bilevel methods as they require neural net parametrizations of the involved functions. Also, these methods were introduced in the context of hyperparameter optimization and it is not clear to us yet how they can be extended to the bilevel problems we considered in this paper (which also include, e.g., few-shot meta-learning, as a special case). But, we thank the reviewer for pointing this out and we have added these methods in our related work discussion (see page 15).
>
> Q2. The key concern about the paper is the lack of rigorous experimentation to show that the proposed method is superior to other bilevel optimization approaches. The paper majorly focuses on the hyper-representation experiments and compares the proposed method to only three baselines (e.g. HOZOG, AID-CG, AID-FP). I believe that the paper should experiment with the algorithm in more diverse settings (e.g. hyperparameter optimization) and compare it to other gradient-based methods (e.g. [2], [3], [5] and various ITD based methods). At the moment, I am not convinced that the proposed algorithm is superior to other bilevel optimizers (both in terms of accuracy and computation it required).
>
> A2: Thanks! As you suggested, we have included hyperparameter optimization experiments in supplementary material. Also, we added the ITD-reverse ([Franceschi 17’]) as a baseline for all hyper-representation and hyperparameter optimization experiments. For few-shot meta-learning experiments, we have added MAML [Finn 17’] and ANIL [Raghu 19’] in the revision as baselines and included results for two different backbone networks. We have not found the source code of STNs for the experiments we considered (e.g., few-shot meta-learning). We will try our best to add more experiments as you suggest.
>
> References:
>
> [Franceschi 17’] Luca Franceschi, Michele Donini, Paolo Frasconi, Massimiliano Pontil. Forward and Reverse Gradient-Based Hyperparameter Optimization. ICML 2017.
>
> [Finn 17’] Chelsea Finn, Pieter Abbeel, and Sergey Levine. Model-agnostic meta-learning for fast adaptation of deep networks. In Proc. International Conference on Machine Learning (ICML), 2017.
>
> [Raghu 19’] Aniruddh Raghu, Maithra Raghu, Samy Bengio, and Oriol Vinyals. Rapid learning or feature reuse? towards understanding the effectiveness of MAML. International Conference on Learning Rep- resentations (ICLR), 2019.
>
>
> Q3: In the first experiment, the authors do not properly explain why there is an improvement in the outer data accuracy. Does this mean that the proposed method approximates the response Jacobian more accurately than CG?
>
> A3: Good question! In fact for the first experiment, except HOZOG (which did not converge for the two-layer net setting), all other compared methods will drive the outer loss down to zero in a longer time horizon. Here, the improvement was in the sense that our method was able to reach a near-zero outer loss at the fastest rate. On a different note, we notice that the AID-CG method exhibits unstable behaviors during the optimization process in this experiment. In fact, AID-CG usually requires more careful choices of step sizes compared to AID-FP or our ESJ method.

---

### Official Review · Reviewer_bgEq · 2021-11-04

**Correctness:** 3
**Technical Novelty And Significance:** 2
**Empirical Novelty And Significance:** 2
**Recommendation:** 5
**Confidence:** 4

**Main Review:**

The paper is generally well written. The core part of this work is the ES Jacobian algorithm, which is specialized in solving bilevel optimization problems. The assumptions made for analyzing the bilevel algorithms are standard, where the main theoretical contributions are Prop.1 and 2 that serve as the pillars of showing the convergence of the bilevel algorithms. I found that method is novel and interesting, but still, have the following concerns regarding the contribution and technical correctness of this work:

1) The proposed ES algorithm is quite similar to the zeroth-order methods. What are the unique differences here? Comparing the first-order and zeroth-order methods, the theoretically guaranteed convergence rate of the zeroth-order ones is slower than the first-order algorithm with a dependency of problem dimension in general. Here, the authors try to argue that the proposed ES is computationally efficient of using the black-box type of methods, which contradicts the common intuition that ES would be slower w.r.t. rate. Please justify.

2) The proposed algorithm is basically a double loop one, since $N$ is a function of $\epsilon$. This again contradicts the intuition that single loop algorithms would be more efficient. To my understanding, the current algorithm cannot be improved as a single loop one.

3) Regarding the convergence rate results, can authors compare the obtained one with existing works, e.g., single- or two-timescale bilevel algorithms? my understanding is that the proposed one for both the deterministic and stochastic cases needs more iterations resulting in higher sample complexity.

4) The required step sizes for the ESJ-S are a little wired. Why do $\alpha,\beta$ be constants? I think the step sizes are either decreasing sequences or depending on $\epsilon$. Please justify.

5) In section 4.3, why not consider MAML and ANIL as baselines?  Also, it seems that all the compared methods have not been converged yet during the training.  Also, it would be great that more statistical measures can be reported such as variance.




**Summary Of The Paper:**

This work mainly focuses on proposing a computational efficient for approximating the response Jacobian matrix in the hypergradient of bilevel optimization with only first-order information of the loss functions. The proposed methods are applicable for developing both deterministic and stochastic bilevel optimization algorithms. The author further show the convergence guarantees and computational complexity of the proposed algorithms. Multiple numerical results showcase the efficiency of the proposed method in applications of machine learning.

**Summary Of The Review:**

Overall, I believe that it is an interesting work, but the proposed is only applicable for solving a class of problems given a special structure of hypergradient. Even the authors justify the computational efficiency of the proposed algorithm numerically w.r.t. running time, the theoretical justification of the efficiency of the iteration and sample complexities should be further addressed.

---

> ### Author Response · Authors · 2021-11-23
> **Thank you for your review**
>
> Many thanks for providing the review! In our revision of the paper, we added new experiments in Figures 1 and 3 in Section 4 and in Appendix D, and made various revisions throughout the paper based on all reviewers’ comments. All our changes are highlighted with blue-colored texts. New comments on these changes are very welcome!
>
> Q1: The proposed ES algorithm is quite similar to the zeroth-order methods. What are the unique differences here? Comparing the first-order and zeroth-order methods, the theoretically guaranteed convergence rate of the zeroth-order ones is slower than the first-order algorithm with a dependency of problem dimension in general. Here, the authors try to argue that the proposed ES is computationally efficient of using the black-box type of methods, which contradicts the common intuition that ES would be slower w.r.t. rate. Please justify.
>
> A1: Good question! Note that the unique challenge of bilevel optimization is that even the first-order (i.e., gradient based) algorithms involve second-order computations. Thus, a high-level view of our method is to use zeroth-order computations to replace second-order computations. More specifically, we use zeroth-order methods to approximate the response Jacobian, which is the major computational bottleneck in estimating the hypergradient. As verified by our experiments, even with the dependence on dimensions, our zeroth-order method is still significantly advantageous compared to the baseline algorithms that need second-order computations.
>
> Q2: The proposed algorithm is basically a double loop one, since N is a function of \epsilon. This again contradicts the intuition that single loop algorithms would be more efficient. To my understanding, the current algorithm cannot be improved as a single loop one.
>
> A2: Thanks! Yes, the proposed algorithm is indeed a double loop one. While the intuition you mentioned is true in conventional optimization problems such as in minimization and minimax optimization, such a rule practice does not hold for bilevel optimization. This is because in bilevel optimization, the outer loop step is usually more expensive (as it involves second-order computations), which suggests that single loop algorithms (resulting in more outer iterations) would be less efficient for bilevel optimization. The theoretical and empirical grounding of this intuition can be also found in [Yang 21’].
>
> [Yang 21’] J. Yang, K. Ji, Y. Liang. "Provably Faster Algorithms for Bilevel Optimization". NeurIPS 2021.
>
> Q3: Regarding the convergence rate results, can authors compare the obtained one with existing works, e.g., single- or two-timescale bilevel algorithms? my understanding is that the proposed one for both the deterministic and stochastic cases needs more iterations resulting in higher sample complexity.
>
> A3: Good point! Theorem 1 shows that our method has higher gradient computations than ITD and AID baselines, but fully eliminates computations of second-order information in the baselines (which is the major computational bottleneck). Since there is no clear theory to tell how many zeroth-order computations is equivalent to how many second-order information computations, we demonstrate by experiments that our method outperforms the existing baselines in computational efficiency.
>
> Q4: The required step sizes for the ESJ-S are a little weird. Why do \alpha and \beta be constants? I think the step sizes are either decreasing sequences or depending on \epsilon. Please justify.
>
> A4: Thanks! In fact there is a trade-off between the selections of stepsizes and batch sizes to achieve the convergence in the stochastic optimization. In our analysis, we choose the batch sizes to be in an order of $1/epsion$ so that the stepsizes can be chosen as constants. On the other hand, if we choose the stepsizes to be decreasing or depending on $\epsilon$, the batch sizes can be relaxed to be a constant level.
>
> Q5: In Section 4.3, why not consider MAML and ANIL  as baselines? Also, it seems that all the compared methods have not been converged yet during the training. Also, it would be great that more statistical measures can be reported such as variance.
>
> A5: In our revision, we have added MAML and ANIL in the baselines in the meta-learning experiment in section 4.3, where it can be seen in Figure 3. that our proposed method is much better. We did not include these two methods previously because their original papers did not include results on the ResNet-12 (they reported only on a smaller network -- 4-layer CNN). Now we included the comparisons both on ResNet-12 and on the 4-layer CNN. Yes, all methods have converged except  MetaOptNet which will converge after 5 days (as reported in their paper). But we did not consider such a very long time horizon as we were more interested in the computational efficiency of the compared methods.

---

### Official Review · Reviewer_Papm · 2021-11-05

**Correctness:** 2
**Technical Novelty And Significance:** 3
**Empirical Novelty And Significance:** 3
**Recommendation:** 5
**Confidence:** 3

**Main Review:**

This paper tackles a very relevant problem. to the machine learning community. It introduces two interesting new algorithms, executing on a simple but elegant idea, which is to limit the black box to its bare minimum, and leverage whatever problem structure of the hypergradient possible.
The writing is clear and enjoyable to read, and the problem is well motivated.

A number of significant issues remain, however.
1. The paper references deep neural networks many times, but crucially relies on the objective being strongly convex with respect to y, which if I understand correctly are the parameters of the model. Deep learning optimisation landscapes are notoriously not even convex, let alone strongly convex. If y is actually only the final layer of a neural network, this needs to be explained right from the start to put the exposition in context. Further, this doesn't seem compatible with the claim that all the 12M resnet12 parameters are learned in the experiment. Some clarity there would go a long way.

2. There are some overclaims, notable regarding 'large' neural networks,  when only fairly small ones are used in the experiments (that is in itself not an issue; claiming that they are large scale when they are not is a problem though).

3. Another overclaim is that the proposed algorithms are 'efficient'. Only requiring gradient computation does not make an algorithm efficient, particularly when it requires NQ *full* gradient steps per outer step. A comprehensive cost analysis with AID and ITD which use second-order information is necessary here, to see when N and Q become too big for this approach to yield gains.

4. The theoretical results, while interesting, are not properly put into context. First, the upper bounds contain a massive additive constant, which isn't smaller than any epsilon. So the results are only valid when this constant is smaller than epsilon, which should be made clear. Second, the settings needed to make the constant small are drastic, meaning Q needs to be in p/epsilon, which if p is the amount of weights of a neural network means using incredibly big Qs, and in turn massive amounts of full gradient steps.
No context is given here on the limitations of the theoretical results, which appear to be all but unusable from a practical perspective.

On a related note, in proposition 3 we need at least some context on how big the constants are.

5.  The experiments are all using small scale neural networks, so claiming that the approach succeeds on large neural nets is misleading.
Finally, there seems to be a discrepancy between the analysis on strongly convex objectives, and the experiments using non-strongly convex objectives.

Question: is it necessary to use the same batch sampling path for every mu? Has this been ablated?

All told, this paper introduces promising new ideas, but at this stage the execution is a bit lacking.  At this stage I do not recommend it for publication at ICLR, but I encourage the authors to take this feedback to heart and improve the paper so that it can have all the more impact on the community.

**Summary Of The Paper:**

The paper introduce two new algorithms to tackle bi-level optimisation, which has wide-ranging application in machine learning.
Crucially, these two algorithms forego second order gradient computation completely, favoring instead an evolution strategies approach.
While this approach has been proposed before in this context, the new algorithms are more efficient because they do not treat the whole problem as a black box, and leverage the structure of the hypergradient.
Theoretical convergence analyses are then provided for both the full gradient and the stochastic gradient alternatives, as well as experiments on various bi-level optimisation problems, showcasing improved performance over previous approaches.

**Summary Of The Review:**

Interesting new methods to tackle a very relevant problem.
Unfortunately a few too many issues remain in the presentation of the paper to make it ready for publication this time around, but it absolutely can be with a bit of additional work.

---

> ### Author Response · Authors · 2021-11-23
> **Continued response...**
>
> Q5: The experiments are all using small scale neural networks, so claiming that the approach succeeds on large neural nets is misleading. Finally, there seems to be a discrepancy between the analysis on strongly convex objectives, and the experiments using non-strongly convex objectives.
>
> A5: See answer A1 and A2. In all our experiments, the inner objective is strongly-convex, but the outer objective can be a general nonconvex function so that our theoretical assumptions on the inner and outer objectives are satisfied.
>
> Q6: is it necessary to use the same batch sampling path for every mu? Has this been ablated?
>
> A6: Good question! We use the same batch sampling path so that the stochastic gradient-free method introduced in [Nesterov 11’] and further elaborated in [Ghadimi 13’] can be applied. This is just for theoretical reasons. We believe different batch sampling paths for different $\mu$ should also perform well in practice.
>
> References:
>
> [Nesterov 11’] Yurii Nesterov and Vladimir Spokoiny. Random gradient-free minimization of convex functions. Foundations of Computational Mathematics, 2011.
>
> [Ghadimi 13’] Saeed Ghadimi and Guanghui Lan. Stochastic first-and zeroth-order methods for nonconvex stochastic programming. SIAM Journal on Optimization, 2013.

---

> ### Author Response · Authors · 2021-11-23
> **Continued response...**
>
> Q3: Another overclaim is that the proposed algorithms are 'efficient'. Only requiring gradient computation does not make an algorithm efficient, particularly when it requires NQ full gradient steps per outer step. A comprehensive cost analysis with AID and ITD which use second-order information is necessary here, to see when N and Q become too big for this approach to yield gains
>
> A3: Our claim of “efficient” is based on experimental evidence provided in the paper, not just “simply removing the Hessian computation”. In experiments, N and Q are hyperparameters that can be chosen properly as long as a good performance is guaranteed. This has been verified by all our experiments that even small N and Q can yield superior performance. We don’t see the reason that the reviewer requires our algorithm to still yield gain for large N and Q.
>
> Regarding the cost analysis, please note that the comprehensive analysis of AID and ITD methods have already been provided in the literature (studied in [Ji 21’], [Grazzi 20’]). The comparison here is clear: our algorithm requires more gradient computations, but without Hessian-vector computations; whereas existing AID and ITD methods require less gradient computations, but have significant Hessian-vector computations (studied in [Ji 21’], [Grazzi 20’]). In theory, we are not aware of any accurate model to tell how many gradient computations are equal to how many Hessian-vector computations. Thus, experiments have the final say about how these methods compare. Our experiments clearly demonstrate the higher efficiency of our algorithm than other bilevel algorithms. Note that in our experiments, we tune hyperparameters of all comparison algorithms to the best for a fair comparison.
>
> References:
>
> [Grazzi 20’]  Riccardo Grazzi, Luca Franceschi, Massimiliano Pontil, and Saverio Salzo. On the Iteration Complexity of Hypergradient Computation. International Conference on Machine Learning (ICML)), 2020.
>
> [Ji 21’] Kayi Ji, Junjie Yang, and Yingbin Liang. Bilevel optimization: Convergence analysis and enhanced
> design. International Conference on Machine Learning (ICML)), 2021.
>
>
> Q4: The theoretical results, while interesting, are not properly put into context. First, the upper bounds contain a massive additive constant, which isn't smaller than any epsilon. So the results are only valid when this constant is smaller than epsilon, which should be made clear. Second, the settings needed to make the constant small are drastic, meaning Q needs to be in p/epsilon, which if p is the amount of weights of a neural network means using incredibly big Qs, and in turn massive amounts of full gradient steps. No context is given here on the limitations of the theoretical results, which appear to be all but unusable from a practical perspective.
>
> A4: We first clarify that it is the convention of the convergence analysis that the constant should be kept at the epsilon level to achieve \epsilon-accurate stationary point. Clearly, by choosing Q and $\mu$ as indicated in Theorem 1, then all terms should be less than \epsilon.
>
> We agree that our theory requires large Qs to guarantee the convergence. But such theory is to establish the worst-case performance guarantee for the proposed method. For many practical experiments, the hyperparameter configurations and setups for the worst-case scenarios are unnecessary to give a good performance. This is why many methods with worst-case performance guarantee will need a careful hyperparameter selection. In fact, the similar dependency of the bounds w.r.t. problem size exists for many famous zeroth-order methods --see, e.g., [Nesterov 11’], [Ghadimi 13’], but their algorithms have been extremely useful in practice such as in adversarial machine learning. All we want to convey is that practitioners don’t need to follow theory exactly to implement an algorithm in the best way, but they do need the theory a lot of times to suggest a convergent algorithm to start with.
>
> References:
>
> [Nesterov 11’] Yurii Nesterov and Vladimir Spokoiny. Random gradient-free minimization of convex functions. Foundations of Computational Mathematics, 2011.
>
> [Ghadimi 13’] Saeed Ghadimi and Guanghui Lan. Stochastic first-and zeroth-order methods for nonconvex stochastic programming. SIAM Journal on Optimization, 2013.

---

> ### Author Response · Authors · 2021-11-23
> **Thank you for your review**
>
> Many thanks for providing the review! In our revision of the paper, we added new experiments in Figures 1 and 3 in Section 4 and in Appendix D, and made various revisions throughout the paper based on all reviewers’ comments. All our changes are highlighted with blue-colored texts. New comments on these changes are very welcome!
>
> Q1: The paper references deep neural networks many times, but crucially relies on the objective being strongly convex with respect to y, which if I understand correctly are the parameters of the model. Deep learning optimization landscapes are notoriously not even convex, let alone strongly convex. If y is actually only the final layer of a neural network, this needs to be explained right from the start to put the exposition in context. Further, this doesn't seem compatible with the claim that all the 12M resnet12 parameters are learned in the experiment. Some clarity there would go a long way.
>
> A1: Good question! For theoretical purposes, the objective of the inner problem is assumed to be strongly-convex, but the outer objective can be a general nonconvex function (see, e.g.,  [Ghadimi 18’], [Ji 21’]). This is a standard and crucial assumption in the analysis of bilevel optimization algorithms. In fact, the nonconvex inner problem setting is an open problem and to the best of our knowledge, it is still not clear yet if one can come up with an algorithm with theoretical convergence guarantee in this case.
>
> For all experiments that we study, the inner loop satisfies strongly convex geometry. For example in few-shot meta-learning experiments, we set the task-specific parameters to be the weights of the last classification layer so that the resulting bilevel problem has a strongly-convex inner problem. The nonconvex outer problem trains the remaining layers (note our theoretical study allows the outer problem to be nonconvex). Such a setting captures many meta-learning problems studied in the literature (see, e.g., [Raghu 19’], [Liu 21’]).
>
> We did learn all 12M parameters. In fact, our hypergradient estimator is not used to learn the task-specific parameters (which are learned with just N steps of gradient descent). Instead, our hypergradient estimator is used to train the remaining convolutional part (i.e., all layers except the last one) of the ResNet12. That is the reason we claim that our estimators were able to learn the 12M weights of the network.
>
> References:
>
> [Ghadimi 18’] Saeed Ghadimi and Mengdi Wang. Approximation methods for bilevel programming. arXiv preprint arXiv:1802.02246, 2018.
> [Raghu 19’] Aniruddh Raghu, Maithra Raghu, Samy Bengio, and Oriol Vinyals. Rapid learning or feature reuse? towards understanding the effectiveness of MAML. ICLR 2019.
>
> [Ji 21’] Kayi Ji, Junjie Yang, and Yingbin Liang. Bilevel optimization: Convergence analysis and enhanced
> design. International Conference on Machine Learning (ICML)), 2021.
>
> [Liu 21’] Risheng Liu, Yaohua Liu,  Shangzhi Zeng, Jin Zhang. Towards Gradient-based Bilevel Optimization with Non-convex Followers and Beyond. NeurIPS 2021.
>
>
> Q2: There are some overclaims, notable regarding 'large' neural networks, when only fairly small ones are used in the experiments (that is in itself not an issue; claiming that they are large scale when they are not is a problem though).
>
> A2: Thanks for bringing this up! In our revision, instead of stating “large neural networks”, we have specified “ResNet-12” to be precise. Please also note that, for few-shot meta-learning, ResNet-12 is already the largest network that has been trained in the literature (see, e.g., [Lee 19’], [Liu 21’]). The networks typically considered in most of few-shot meta-learning literature such as in (MAML [Finn 17’] and ANIL [Rajeswaran 19’]) are much smaller (e.g., 4-layer CNN for miniImageNet dataset and OmniglotNet for Omniglot dataset). And all experiments in all bilevel studies in the literature train much smaller networks as well. The reason for this is due to the nested hierarchy of the problem, which makes it hard to scale to the extremely large networks employed in other machine learning problems.
>
> References:
>
> [Finn 17’] Chelsea Finn, Pieter Abbeel, and Sergey Levine. Model-agnostic meta-learning for fast adaptation of deep networks. In Proc. International Conference on Machine Learning (ICML), 2017.
>
> [Raghu 19’] Aniruddh Raghu, Maithra Raghu, Samy Bengio, and Oriol Vinyals. Rapid learning or feature reuse? towards understanding the effectiveness of MAML. International Conference on Learning Rep- resentations (ICLR), 2019.
>
> [Lee 19’] Kwonjoon Lee, Subhransu Maji, Avinash Ravichandran, and Stefano Soatto.  Meta-learning with
> differentiable convex optimization.  In Proceedings of the IEEE/CVF Conference on Computer Vision and Pattern Recognition, 2019.
>
> [Liu 21’] Risheng Liu, Yaohua Liu,  Shangzhi Zeng, Jin Zhang. Towards Gradient-based Bilevel Optimization with Non-convex Followers and Beyond. NeurIPS 2021.

---

### Decision · Program_Chairs · 2022-01-20

**Decision:**

Reject

**Comment:**

This paper presents an algorithm for approximating the hypergradient for bilevel optimization using a trick based on evolution strategies. It seems like an interesting approach, somewhat reminiscent of STNs, so it's interesting to see experiments with it.

I have a big concern about the proposed justification of the method, namely that each iteration is more efficient than methods based on HVPs. The authors claim that because they only require gradient computations and not HVPs, their method is more efficient. However, as various reviewers point out, the proposed method requires numerous inner optimization runs. By comparison, a method based on unrolled backprop simply requires a single inner run, followed by backprop on the trajectory; hence it should be about as expensive as 2-3 inner optimizations (or less if it is truncated BPTT). Similarly, each HVP has a small multiple of the cost of an inner optimization step, so methods based on HVPs ought to be cheaper unless they're doing quite a lot of HVPs.

It's conceivable the proposed method could be more efficient than AID, etc. if each hypergradient estimate is more accurate. However, this isn't shown, and it would seem surprising for an ES-based approximation to be more accurate than the exact gradient.

The authors claim in the rebuttal that the efficiency claims aren't based on the theoretical analysis, but rather on the experiments (which use Q=1); however, Section 3.2 still finishes with the conclusion that ESJ is more efficient, which seems problematic.

I encourage the authors to formulate their theoretical claims more carefully and to consider the reviewers' other feedback, and I think this could make an interesting submission for the next cycle.